# SuperMAN: Interpretable and Expressive Networks over Temporally Sparse Heterogeneous Data

**Andrea Zerio**[2],[*]  **Maya Bechler-Speicher**[1][*]

**Maor Huri**[7],  **Marie Vibeke Vestergaard**[2],

**Ran Gilad-Bachrach**[6],  **Tine Jess**[2,3],  **Samir Bhatt**[4,5],  **Aleksejs Sazonovs**[2]

[1]Meta
[2]Center of Excellence for Molecular Prediction of IBD (PREDICT),
 Department of Clinical Medicine, Aalborg University
[3]Department of Gastroenterology & Hepatology, Aalborg University Hospital
[4]University of Copenhagen
[5]Imperial College London
[6]Department of Biomedical Engineering, Tel-Aviv University
[7]Sagol School of Neuroscience, Tel-Aviv University

## Abstract

Real-world temporal data often consists of multiple signal types recorded at irregular, asynchronous intervals. For instance, in the medical domain, different types of blood tests can be measured at different times and frequencies, resulting in fragmented and unevenly scattered temporal data. Similar issues of irregular sampling occur in other domains, such as the monitoring of large systems using event log files. Effectively learning from such data requires handling sets of temporally sparse and heterogeneous signals. In this work, we propose Super Mixing Additive Networks (SuperMAN), a novel and interpretable-by-design framework for learning directly from such heterogeneous signals, by modeling them as sets of implicit graphs. SuperMAN provides diverse interpretability capabilities, including node-level, graph-level, and subset-level importance, and enables practitioners to trade finer-grained interpretability for greater expressivity when domain priors are available. SuperMAN achieves state-of-the-art performance in real-world high-stakes tasks, including predicting Crohn's disease onset and hospital length of stay from routine blood test measurements and detecting fake news. Furthermore, we demonstrate how SuperMAN's interpretability properties assist in revealing disease development phase transitions and provide crucial insights in the healthcare domain.

## 1 Introduction

Modern clinical data consist of diverse types of signals, often collected at irregular and asynchronous time intervals. For example, a patient's medical record may include a set of blood tests taken over their lifetime, where each type of test is performed at its own frequency. As a result, the data can be viewed as a set of sparse temporal signals, each with its own fragmented temporal structure. Similar patterns occur in other domains. For instance, the spread of news articles in social media networks often follow asynchronous tree-like patterns of dissemination. Another example is system event logs, which typically include different types of events occurring at varying times and rates.

---

[1]These authors contributed equally to this work.

A common approach for learning from such data is to align the signals to a fixed-size time grid, thereby enforcing a shared timeline. This is typically achieved by trimming or aggregating signals and filling in missing values through interpolation or learned imputation models (Cao et al., 2018; Tashiro et al., 2021; Wu et al., 2022; Du et al., 2023). However, these procedures can lead to substantial information loss and ignore the informative patterns in the irregularity itself, such as the varying time intervals between different measurement types.

In this work, we introduce Super Mixing Additive Networks (SUPERMAN), a novel framework designed to learn directly from heterogeneous, temporally sparse and irregular signals, without information loss or imputation. SUPERMAN models these signals as sets of implicit graphs by introducing a distance function between each pair of signals within each type of signal. For example, in blood test data, each biomarker type can be modeled as a directed graph whose nodes correspond to individual measurements, with the distance between each pair of measurements being the time delta between them. These implicit graphs may differ in structure, size, and feature space, reflecting the diversity of real-world heterogeneous signals.

SUPERMAN allows for multi-resolution interpretability capabilities, including node-level, graph-level, and subset-level importance. It also enables practitioners to trade fine-grained interpretability for greater expressivity when domain priors are available. Specifically, signal graphs can be grouped into subsets, allowing for more expressive modeling of non-linear interactions within the subsets. This shifts interpretability from individual nodes or graphs to the subset level. SUPERMAN builds on Graph Neural Additive Networks (GNAN) (Bechler-Speicher et al., 2024), an interpretable class of GNNs that operate on individual nodes or graphs. In contrast, SUPERMAN is expressly designed for *sets of implicit graphs* and introduces a graph-grouping mechanism that aggregates signals within subsets of graphs while preserving an additive decomposition across subsets. Rather than applying GNAN directly, SUPERMAN employs an extended and more flexible variant, which we denote by ExtGNAN. Within each graph, ExtGNAN generalises GNAN's univariate feature-shape networks to multivariate feature groups, allowing non-linear dependencies among related features while retaining additive transparency at the group level. In analogy to signal grouping, ExtGNAN supports grouping features into subsets, thereby replacing feature-level interpretability with subset-level interpretability, such as importance scores. We prove that grouping signals or features strictly increases expressivity, and that SUPERMAN is strictly more expressive than GNAN.

We demonstrate the effectiveness of SUPERMAN on real-world high-stakes medical datasets, where it achieves state-of-the-art (SoTA) performance while also providing valuable clinical and biological insights through its interpretability. In addition, we show that SUPERMAN attains SoTA performance in fake news detection, highlighting its flexibility in operating on sets of graphs with arbitrary structure. This contrasts with existing approaches, which are restricted to sets of path-like signals and offer no interpretability. Finally, we demonstrate how SUPERMAN's interpretability reveals phase transitions and provides crucial insights in healthcare.

Our main contributions are as follows:

1. We introduce SUPERMAN, a novel framework for learning directly from sets of sparse, irregular temporal signals, without information loss or imputation.

2. We allow practitioners to integrate domain priors, when available, by grouping features or signal types into subsets. This shifts interpretability from fine-grained (node- or feature-level) to subset-level, while strictly improving expressivity. This capability is particularly valuable in the medical domain, where such priors are common.

3. We provide a theoretical analysis proving that groupings of features and signals make SUPERMAN strictly more expressive.

4. We demonstrate the effectiveness of SUPERMAN on real-world high-stakes medical datasets and fake news detection, achieving state-of-the-art performance in both domains.

5. We show that SUPERMAN provides valuable interpretability capabilities, including node-level, graph-level, and subset-level importance, which yield meaningful clinical and biological insights.

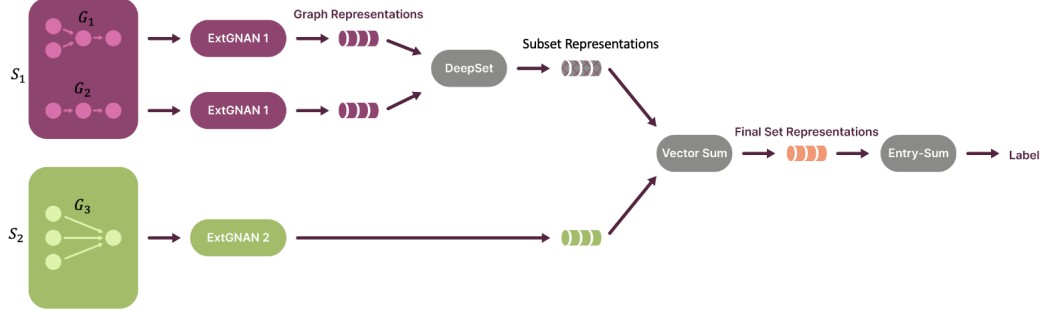

Figure 1: In this example, the input is a set of three graphs, $G_1, G_2, G_3$, grouped into two subsets $S_1$ and $S_2$. Within each subset, the same ExtGNAN instance is applied to all graphs to produce their graph-level representations. For subsets containing multiple graphs, a DeepSets module aggregates these graph representations into a single subset representation. For subsets of size one, the subset representation is simply the graph representation itself. The final set representation is then obtained by summing the subset representations, and the final label prediction is produced by summing the entries of this set representation.

## 2 RELATED WORK

**Graph Neural Additive Networks**   Graph Neural Networks (GNNs) (Kipf & Welling, 2016; Gilmer et al., 2017; Velickovic et al., 2017; Xu et al., 2018) have become the dominant framework for learning over graph-structured data, enabling flexible representation learning across diverse domains such as healthcare (Paul et al., 2024; Ochoa & Mustafa, 2022; Peng et al., 2023), chemistry (Reiser et al., 2022; Jumper et al., 2021) and social networks (Li et al., 2023; Sharma et al., 2024), among others. GNNs leverage both the graph topology and node features to compute learned representations for individual nodes or for entire graphs. Recently, Bechler-Speicher et al. (2024) introduced Graph Neural Additive Networks (GNANs), a novel interpretable-by-design graph learning framework inspired by generalized additive models (GAMs) (Hastie & Tibshirani, 1986; 1987). GNAN applies univariate neural networks to each feature of the nodes separately, and then linearly combines their outputs across nodes to produce node-level and graph-level representations. As features are not mixed non-linearly, GNAN is fully interpretable, and provides feature-level and node-level interpretability which shows exactly how each feature and each node contributes to the final target variable.

**Learning from sparse data**   In many real-world settings, data often presents missing values from irregular sampling and variable feature availability. Recurrent models (Cao et al., 2018) treat missing data as latent variables, while attention-based methods (Du et al., 2023; Wu et al., 2022; Tipirneni & Reddy, 2022; Labach et al., 2023) reconstruct them via contextual masking and temporal blocks. Diffusion models (Tashiro et al., 2021; Alcaraz & Strodthoff, 2022; Senane et al., 2024; Dai et al., 2024) learn conditional distributions over missing values using score-based processes. Graph-based approaches use GNNs to model feature dependencies through bipartite graphs, adaptive message passing, or spatio-temporal attention (You et al., 2020; Cini et al., 2022; Marisca et al., 2022; Ye et al., 2021; Chen et al., 2024). These imputation methods often distort dynamics and may not improve prediction (Qian et al., 2025). An alternative is to model sparsity directly. Neural and Latent ODEs (Chen et al., 2018; Rubanova et al., 2019) address irregular gaps via continuous dynamics but are compute-intensive and rely on missingness encodings. Recent models (Zhang et al., 2021) have used graph representations to capture sparsity without imputation. Importantly, none of the aforementioned methods offer built-in multi-grain interpretability in the way that SUPERMAN does.

## 3 SUPER MIXING ADDITIVE NETWORKS

In this section, we present SUPERMAN, an interpretable and flexible method for learning over sets of arbitrary graphs.

**Preliminaries** SUPERMAN acts on a set of $m$ graphs $S = \{G_1, \ldots, G_m\}$, which can be directed or undirected. Each node $v \in G_i$, $1 \le i \le m$ is associated with a feature vector $x_v \in \mathbb{R}^d$ and a time-stamp $t_v$. SUPERMAN utilizes the distances between each pair of nodes in each graph, denoted by $\Delta_{uv}$, where

$$\Delta_{uv} = \begin{cases} t_u - t_v, & \text{if there exists a path from } u \text{ to } v \\ 0, & \text{otherwise.} \end{cases}$$

Graphs may be provided directly through an explicit layout, e.g., the news propagation graphs we evaluate SUPERMAN over in section 4. More commonly, however, sparse temporal heterogeneous data does not come with a predefined graph structure. In such cases, we construct directed path-graphs for each signal type based on its measurement time stamps, as demonstrated for the medical dataset in section 4. For instance, in patient blood test records, each graph corresponds to a specific biomarker. Nodes in each graph represent individual measurements of that biomarker, annotated with the observed test value as a feature and the associated time stamp. We mark vectors with bold, and denote the entry $c$ of a vector $\mathbf{h}$ by $[\mathbf{h}]_c$, and the set of entries corresponding to a set of features $S$ by $[\mathbf{h}]_S$.

**Signal Grouping** To incorporate domain priors, the graphs in $S$ can be partitioned into $k$ disjoint subsets $S_1, \ldots, S_k$ with $\bigcup_{i=1}^k S_i = S$. If at least one subset $S'$ contains multiple graphs, the model's expressivity increases, as we prove in Theorem 3.2, at the cost of shifting interpretability from individual nodes and graphs in $S'$ to the subset level. In practice, this means we can attribute importance to $S'$ as a whole, but not to its individual components. This trade-off—enhanced expressivity at reduced granularity of interpretability is especially valuable in domains such as medicine, where priors often suggest natural groupings of signals and interpretability is only needed at the subset level. This is demonstrated in Section 4.

SUPERMAN linearly aggregates representations of the subsets of $S$ to form a final set representation, and then assigns a single label to $S$.

First, SUPERMAN applies a function $\Phi_i$ to each subset $S_i$ to obtain a representation of the subset $S_i$, denoted as $\mathbf{h}_i \in \mathbb{R}^d$.

$$\mathbf{h}_i = \Phi_i(S_i),$$

Then, it produces a representation for the whole set, $\mathbf{h}_S$ by summing the subsets' $\mathbf{h}_S = \sum_{i=1}^k h_i$. Finally, to produce the label, it sums over the $d$ entries of $\mathbf{h}_S$. Overall:

$$\text{SUPERMAN}(S) = \sum_{c=1}^d \sum_{i=1}^k [\Phi_i(S_i)]_c \tag{1}$$

Where $\Phi_i(S_i) = \mathbf{h}_{S_i}$ is a representation of the subset $S_i$.

For subsets of size one, $\Phi_i(S_i)$ applies an Extended GNAN (EXTGNAN), as described in Section 3.1. For subsets containing multiple graphs, a featuregroupgnan is applied to each graph, followed by a DeepSet aggregation (Zaheer et al., 2018) over the resulting vectors. Importantly, each subset is assigned its own EXTGNAN, and all graphs within a subset share the same one. A DeepSet first applies a neural network (NN) $f : \mathbb{R}^d \to \mathbb{R}^d$ for each vector in the set $\{h_l\}_{G_l \in S_i}$, sums the results, and then applies another NN $g : \mathbb{R}^d \to \mathbb{R}^d$.

$$g \left( \sum_{i \in S_2} f(h_i) \right)$$

Here, $g$ and $f$ are NNs of arbitrary depth and width. A high-level visual overview of SUPERMAN is presented in Figure 1. We now turn to define EXTGNAN.

### 3.1 EXTGNAN

In GNAN, univariate NNs are applied to each feature of each node in isolation, to learn a representation for a graph. This has the benefit of generating interpretable models as features do not mix non-linearly. Nonetheless, when interactions between features are crucial for the task, or feature-level interpretability is not required for all features, it may result in sub-par performance. Therefore, EXTGNAN extends GNAN by allowing multivariate NNs to operate on groups of features to gain accuracy at the cost of reducing the feature-level interpretability only for features that are grouped together, and obtaining interpretability for their subset as a whole instead.

Assume that the features are partitioned into $K$ subsets $\{F_l\}_{l=1}^K$. For any subset of features greater than one, EXTGNAN applies a multivariate NN for all the features in the subset together, instead of a univariate NN for each one separately. To learn a representation of a graph $G$, EXTGNAN first computes representations for the nodes of $G$ as follows.

EXTGNAN learns a distance function $\rho(x; \theta) : \mathbb{R} \to \mathbb{R}$ and a set of feature shape functions $\{\psi_l\}_{l=1}^K, \psi_l(X; \theta_k) : \mathbb{R}^{|F_l|} \to \mathbb{R}^{|F_l|}$. Each of these functions is a NN of arbitrary depth. For brevity, we omit the parameterization $\theta$ and $\theta_k$ for the remainder of this section.

The entries of the representation of node $j$ corresponding to the indices of the features in $F_l$, denoted as $[\mathbf{h}_j]_{F_l}$, is computed by summing the contributions of the features in the subset $F_l$ from all nodes in the graph:

$$[\mathbf{h}_j]_{F_l} = \sum_{w \in V} \rho\left(\Delta(w, j)\right) \cdot \psi_l\left([\mathbf{X}_w]_{F_l}\right),$$

where $\Delta(w, j) = t_w - t_j$ and $[\mathbf{X}_w]_{F_l}$ are the features of node $w$ corresponding to the subset $F_l$.

Overall, the full representation of node $j$ can be written as:

$$\mathbf{h}_j = \left([\mathbf{h}_j]_{F_1}, [\mathbf{h}_j]_{F_2}, \ldots, [\mathbf{h}_j]_{F_K}\right).$$

Then EXTGNAN produces a graph representation by summing the node representations,

$$\mathbf{h}_G = \sum_{i \in V} \mathbf{h}_i. \tag{2}$$

This concludes the description of EXTGNAN, which computes graph-level representations. We provide a complexity analysis of SUPERMAN in the appendix, where we also discuss how $\rho$ can be masked to adapt to any complexity limitation, including a linear one. Next, we describe how SUPERMAN combines these representations across sets and enables multi-level interpretability.

### 3.2 NODE, GRAPH AND SUBSET IMPORTANCE

SUPERMAN retains all interpretability properties of GNAN, including feature-level and node-level importance. However, it extends beyond GNAN by operating on sets of graphs rather than single graphs, enabling additional forms of interpretability such as graph-level and subset-level importance. Because SUPERMAN allows a flexible trade-off between interpretability and expressivity, permitting non-linear mixing within graph subsets, some adaptations are required to obtain importance scores. A key property of SUPERMAN's interpretability is that its importance scores directly reflect the contribution of each node, graph, or subset to the predicted label, since these terms are combined additively to produce the final output. Section 4 illustrates how node-level importances yield insightful real-world insights.

We can extract the total contribution of each node $j$ to the prediction by summing the contributions of the node across all feature subsets. This is only valid when the node belongs to a graph that is not combined non-linearly with other graphs, i.e., it belongs to a subset of size one.

Therefore, the contribution of node $j$ is

$$\text{TotalContribution}(j) = \sum_{l=1}^K [\mathbf{h}_j]_{F_k} = \sum_{w \in V} \rho\left(\Delta(w, j)\right) \sum_{l=1}^K \psi_k\left([\mathbf{x}_w]_l, l \in F_k\right). \tag{3}$$

The contribution of a graph $G$ is then

$$\text{TotalContribution}(G) = \sum_{v \in G} \text{TotalContribution}(v).$$

For graphs that are mixed non-linearly, i.e., graphs that belong in subsets of size greater than one, we provide instead the total contribution of the set to the final prediction

$$\text{TotalContribution}(S) = \sum_{l=1}^{K} [\mathbf{S}]_{F_k}. \tag{4}$$

### 3.3 Expressivity properties

In this section, we provide a theoretical analysis of the expressiveness of SUPERMAN. Proofs are provided in the Appendix

**Theorem 3.1.** SUPERMAN *is strictly more expressive than GNAN.*

The following theorem shows that a SUPERMAN which is applied to subsets of graphs of size at least two, is more expressive than a SUPERMAN that is applied to only subsets of size one:

**Theorem 3.2.** *Let $S$ be a set of graphs $\{G_i\}_{j=1}^{m}$. Let $S_1 = \{S_i\}_{i=1}^{m}$ be a partition of $S$ such that $|S_i| = 1$. Let $S_2 = \{S_i\}_{i=1}^{k}$ such that there exists $k$ with $|S_k| > 1$. with a subset partition $\{S_i\}_{i=1}^{k}$. Then a SUPERMAN trained over $S_2$ is strictly more expressive than a SUPERMAN trained over $S_1$.*

## 4 Empirical evaluation

In this section, we evaluate SUPERMAN on real-world tasks, and demonstrate its interpretability properties [1].

### 4.1 Medical predictions

We evaluate SUPERMAN on two high-impact clinical prediction tasks: LoS of intensive care patients and onset of CD. In both settings, each individual is represented as a set of time-stamped biomarker trajectories, where each trajectory forms a directed path graph with nodes corresponding to test results and edges encoding the time elapsed between measurements. This representation preserves the temporal structure of each biomarker independently while enabling joint reasoning across biomarkers during learning.

**Data**  Next, we describe the two medical datasets used in our evaluation.

*P12 (ICU Length of Stay):* The PhysioNet2012 (P12) dataset, introduced by Goldberger et al. (2000), contains records from 11,988 intensive-care unit (ICU) patients, following the exclusion of 12 samples deemed inappropriate according to the criteria in Horn et al. (2020). For each patient, longitudinal measurements from 36 physiological signals were recorded over the initial 48 hours of ICU admission. Additionally, each patient has a static profile comprising 9 features, including demographic and clinical attributes such as age and gender. The dataset is labelled for a binary classification task: predicting whether or not the total LoS in the ICU exceeded 72 hours. The dataset is highly imbalanced, with ∼93% positive samples. To balance the classes we perform batch minority class upsampling, following the example in Zhang et al. (2021).

*Crohn's Disease (CD Onset):* The Danish health registries (DHR) are comprehensive, nationwide databases covering healthcare interactions for over 9.5 million individuals (Pedersen, 2011). A key resource is the Registry of Laboratory Results for Research (RLRR), which has collected laboratory test results from hospitals and general practitioners since 2015 (Arendt et al., 2020). We first identify 8,567 individuals in the DHR who are later diagnosed with CD and use them as our patient class. We then construct a control pool by randomly sampling individuals from the DHR and downsampling by age to match the expected frequency of blood tests in the prediagnostic period, reflecting that CD

---

[1] Implementation is provided in
https://github.com/azerio/Super-Mixing-Additive-Networks---SuperMAN

typically manifests in early adulthood ( 20–30 years). From this age-matched control pool, we sample 8,567 controls, yielding two balanced classes. For each person, we extracted temporal trajectories of 17 routinely measured biomarkers, reflecting key physiological processes. The complete list and descriptions of these biomarkers are provided in the Appendix. The task is binary classification: predicting future CD onset from pre-diagnostic medical histories.

**Setup**   We compare SUPERMAN to 8 baselines from Zhang et al. (2021), spanning both sequential and graph-based models, including: *Transformer* (Vaswani et al., 2017), a vanilla self-attention model applied to irregular time series; *Trans-mean*, which combines a Transformer with mean imputation of missing values; *GRU-D* (Che et al., 2016), a gated recurrent model with decay terms that encode informative missingness; *SeFT* (Horn et al., 2020), which treats each record as a set of timestamped feature-value pairs and aggregates them with permutation-invariant encoders; *mTAND* (Shukla & Marlin, 2021), a multi-time attention architecture that outperforms a broad

Table 1: Evaluation of SUPERMAN on two real-world medical tasks. The metric reported is the mean AUPRC with standard deviation, calculated over 3 random seeds.

| Methods | LoS in ICU | CD onset |
|---|---|---|
| Transformer | $96.06 \pm 0.32$ | $75.60 \pm 0.52$ |
| Trans-mean | $96.44 \pm 0.17$ | $75.96 \pm 0.92$ |
| GRU-D | $95.91 \pm 2.10$ | $83.36 \pm 0.40$ |
| SeFT | $95.89 \pm 0.08$ | $71.22 \pm 2.30$ |
| mTAND | $93.02 \pm 1.04$ | $83.17 \pm 0.67$ |
| $DGM^2$ | $97.00 \pm 0.40$ | $83.02 \pm 0.56$ |
| MTGNN | $96.20 \pm 0.78$ | $75.26 \pm 3.04$ |
| RAINDROP | $96.32 \pm 0.13$ | $82.60 \pm 0.82$ |
| SUPERMAN | $\mathbf{97.41 \pm 0.38}$ | $\mathbf{83.93 \pm 0.27}$ |

range of RNN- and ODE-based models on irregular data; $DGM^2$ (Wu et al., 2021) and *MTGNN* (Wu et al., 2020), graph-based methods originally proposed for multivariate time-series forecasting; and *Raindrop* (Zhang et al., 2021), a state-of-the-art graph model for sparse, irregular EHR time series. For the $P12$ we used the splits as in (Zhang et al., 2021). For CD, we randomly split the data into train (80%), validation (10%), and test (10%) sets. We conducted a grid search by training on the training set and evaluating on the validation set. We then selected the best-performing model over the validation set and report results over the test set. We define each individual biomarker as a unique signal type and group them according to coarse physiological categories. Importantly, these clinically-guided groups are based on broad domain knowledge and do not rely on specialized expertise. We also examine data-driven grouping that does not rely on any priors, based on the insights presented in Subsection 4.1.1. Due to space limitations inTable 4, App. F. We tune the biomarker subsets over no grouping at all (i.e, full interpretability), and 5 additional subset groupings motivated by public common clinical knowledge. To account for class imbalance, and as commonly done for these datasets, we report the average AUPRC score and standard-deviation of the selected configuration with 3 random seeds. Additional details on the datasets, experimental setup, subset groupings, and hyperparameter configurations are provided in the Appendix. We also present Calibration and robustness evaluations in Appendix G.

**Results**   The results in Table 1 show that SUPERMAN achieves the highest AUPRC across both medical prediction tasks. On Length of Stay in ICU, it improves upon the best baseline with a relative uplift of about $0.41$ points, while on Crohn's Disease onset, it outperforms the best baseline by $0.57$ points. Notably, these improvements are achieved while SUPERMAN also provides interpretability properties, as shown in detail in the next subsection.

### 4.1.1    CLINICAL INSIGHTS THROUGH INTERPRETABILITY

Beyond predictive performance, a central strength of SUPERMAN is that interpretability is built into the model by design, rather than added post hoc. This allows for fine-grained, temporally resolved explanations that go beyond global feature importance, enabling users to understand how and when specific biomarkers influence the model's predictions. In high-stakes domains such as healthcare, this level of transparency is crucial. Clinical decision-making often depends not only on the outcome of a prediction but on a clear understanding of the reasoning behind it. Models that can provide such insights are far more likely to be trusted, audited, and integrated into clinical workflows. To highlight the interpretability of SUPERMAN, we conduct attribution analyses on both the CD and P12 datasets, examining how the model assigns importance across time and signals.

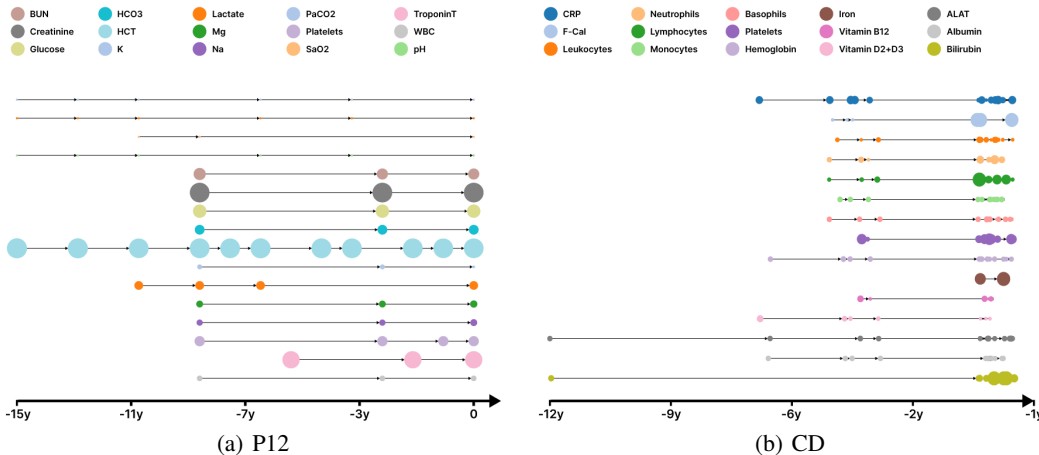

Figure 2: Node-level importances for two individuals from (a) the P12 ICU LoS and (b) CD onset datasets. Node size indicates the exact node (measurement) contribution to the prediction.

**Critical phase detection through node-level importance**    In the SUPERMAN framework, nodes represent individual signals within a biomarker trajectory. As such, highly influential nodes can highlight critical phases where specific signals most strongly impact the prediction. We use Equation (3) to quantify node-level contributions across biomarkers in both the CD and P12 datasets. To comply with privacy and data protection legislation, CD data is anonymised via noise and temporal shifting. Figure 2 displays node-level importance across biomarker trajectories for two randomly selected individuals from each dataset, with node size reflecting the magnitude of each node's contribution to the model's prediction. In both clinical tasks, the model's attributions appear consistent with established biomedical knowledge. For CD prediction, SUPERMAN highlights key inflammatory and immune markers, such as F-Cal, platelets, and lymphocytes, as primary contributors. All of these are known to play central roles in disease onset (Vestergaard et al., 2023). In the P12 example, the model assigns high importance to markers of renal function, liver injury, cardiac stress, and metabolic imbalance, aligning well with clinical predictors of severity in intensive care settings.

**Total biomarker contribution**    In addition to node-level importance scores, SUPERMAN also provides subset-level scores, quantifying the contribution of entire biomarker groups to the model's prediction. This enables flexible, system-level interpretability, allowing users to assess the collective influence of physiologically related biomarkers on the target variable. We conduct a subset-level importance analysis on the CD dataset using two clinically motivated grouping strategies. In the first, we assess individual biomarkers to determine whether the model prioritizes features known to be associated with CD onset, allowing us to verify its alignment with established biomedical knowledge. In the second, we group biomarkers into clinically coherent subsets based on physiological function, such as immune response; inflammation; oxygen transport; and liver function; to examine whether the model captures system-level patterns consistent with disease progression. Full details on the clinical significance of these groupings are provided in the Appendix.

For each grouping strategy, we trained a separate instance of SUPERMAN. To quantify the importance of each biomarker subset, we use Equation (4), measuring how the model's prediction changes when increasing the noise added to the features of nodes within that subset. To introduce noise in a structured way, we apply PCA (Abdi & Williams, 2010) to the feature vectors of all nodes corresponding to biomarkers in the group, and progressively perturb the input along the 1st principal component. This procedure is performed independently for each subset and allows us to assess the model's sensitivity to perturbations in biologically meaningful groupings, offering insight into the relative predictive weight of each group. Importantly, SUPERMAN is interpretable by design, which makes perturbation analysis directly reflect its internal computation. Unlike post-hoc attribution, we do not estimate influence indirectly but observe how the additive contribution changes. Adding structured noise (via PCA) and measuring the resulting prediction shift faithfully traces the model's internal

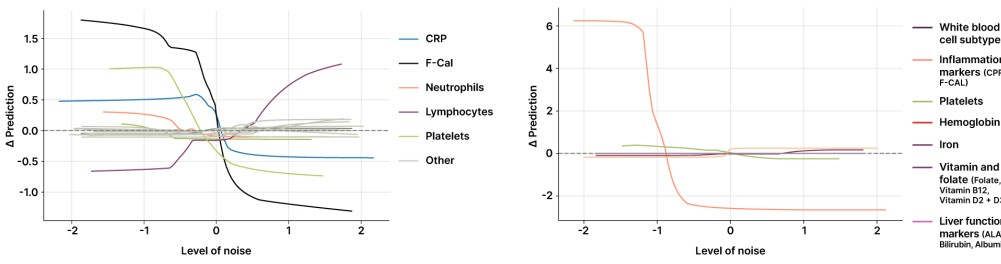

(a) Single-biomarker groups SUPERMAN      (b) Best-performing SUPERMAN groups

Figure 3: Subset-level contribution curves for Crohn's Disease prediction. Each curve shows how the SUPERMAN's output changes as increasing noise is added to the latent representation of a biomarker group. (a) uses individual biomarkers; (b) uses physiologically coherent groups.

mechanism rather than relying on an external approximation. Thus, perturbation effects align exactly with feature importance, making the experiment both natural and principled. Figure 3 presents the results of the subset-level attribution analysis. In the single-biomarker setting (Figure 3(a)), F-Cal, platelets, and lymphocytes show strong directional effects on model output. F-Cal and platelets are positively associated with CD risk, while lymphocytes have an inverse effect, findings that are both biologically grounded and consistent with prior work (Vestergaard et al., 2023). In the clinically-coherent group setting (Figure 3(b)), the inflammation subset emerges as the most influential, with a pronounced non-linear effect on predictions, aligning with established diagnostic relevance in CD (Vestergaard et al., 2023). We note that the above also serves as a quantitative measure of faithfulness, as in SUPERMAN, interpretability is not post-hoc but built into the model architecture by design. Specifically, the contributions of nodes, graphs, or subsets are explicitly and additively used in computing the final prediction. This means that importance scores correspond directly to the actual values that are summed to produce the output label, making them faithful by construction.

## 4.2 FAKE-NEWS DETECTION

**Data** GossipCop (GOS) is a dataset of news articles annotated by professional journalists and fact-checkers, containing both content-based labels and social context information verified through the GossipCop fact-checking platform. It is composed of 5,464 tree-structured graphs based on sharing information, where the news article is the root node and sharing users are subsequent nodes in the cascade, with edges signifying sharing relationships. Each node in these graphs is associated with 4 features: 768-dimensional embeddings generated using a pretrained BERT model on user historical posts, 300-dimensional embeddings from a pretrained word2vec (Mikolov et al., 2013) on the same historical posts, 10-dimensional features extracted from user profiles, and a 310-dimensional

Table 2: Evaluation of SUPERMAN on the Gossipcop (GOS) fake news detection dataset.

| Methods | Accuracy |
|---|---|
| GATv2 | 96.10 ± 0.3 |
| GraphConv | 96.77 ± 0.1 |
| GraphSage | 94.45 ± 1.5 |
| GCNFN | 96.52 ± 0.2 |
| SUPERMAN | **97.34 ± 0.2** |

"content" feature that combines the 300-dimensional embedding of user comments with the 10-dimensional profile features. Each graph is labelled to indicate whether it originates from a fake news post or not. We decomposed the tree into a set of directed graphs rooted at the origin, each representing a distinct path of news propagation.

**Setup** While SUPERMAN can be applied to sets of graphs with any structure, the baselines used in the CD and P12 experiments are limited to path-like graphs and are unable to act on more complex graphs as in this dataset. Therefore, we instead evaluate SUPERMAN against 4 GNNs, including GATv2 (Brody et al., 2022), GraphConv (Morris et al., 2021), GraphSAGE (Hamilton et al., 2018) and GCNFN (Monti et al., 2019). We use random splits of train (80%), validation (10%), and test (10%) sets over the data and selected the best performing model over the validation set. We then report the average Accuracy score and std of the selected configuration with three seeds. Since many

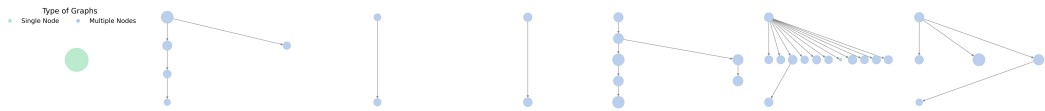

Figure 4: Node importance for fake-news spread graphs, over the GOS dataset. The node size corresponds to its importance learned by SUPERMAN, according to Equation (3). All graphs with a single node are grouped into one subset. Therefore, the importance is provided on the subset level rather than the node level (green node).

subgraphs reduce to a single node after decomposition, we group all size-one graphs into a shared subset and combine them non-linearly. For the remaining graphs, whose identities are not uniquely distinguishable, we apply a shared EXTGNAN. In total, we use two distinct EXTGNAN instances for this experiment.

**Results and node-level importance**    Results are provided in Table 2. SUPERMAN outperforms all baselines, with an uplift of 0.57 accuracy points. Figure 4 presents the node importance of a random sample from the datasets, where the size of a node corresponds to its importance score according to Equation (3).

### 4.2.1    ABLATION STUDY

Finally, we carry out an ablation study on the CD dataset, to isolate the impact of key components on performance. Specifically, we take the best-performing configuration from the training-data grid search and re-train it with individual components ablated to assess their effect on performance. We test: (i) replacing DeepSet with mean pooling to assess the value of learned non-linear aggregation; (ii) replacing the distance function NN ($\rho$) with a constant 1 value.

Table 3: Ablation study of SuperMAN components.

| Ablation | AUPRC drop |
|---|---|
| (i) DeepSet→mean pooling | *- 19.98% $\pm$ .28 %* |
| (ii) $\rho$→1 | *- 12.39% $\pm$ 1.39%* |
| (iii) ExtGNAN→MLP | *- 15.00% $\pm$ 2.09 %* |
| (iv) ExtGNAN→Identity | *- 17.70% $\pm$ 0.15%* |
| (v) ExtGNAN→GNAN | *- 4.38 % $\pm$ 2.85%* |

This tests the usefulness of structural information; (iii) substituting ExtGNAN with a node-wise MLP to test the importance of graph inductive bias; (iv) replacing ExtGNAN with an identity mapping as a lower bound without feature learning; and (v) using standard GNAN (no multivariate feature groups) to evaluate the benefit of grouped feature processing. We report the performance degradation (AUPRC difference) in Table 3. The ablation results demonstrate that the core components of SUPERMAN are critical to its effectiveness, as their removal consistently leads to notable performance degradation.

## 5    CONCLUSION

We introduced Super Mixing Additive Networks (SUPERMAN), a framework for learning from sets of graphs that represent irregular and asynchronous temporal signals. SUPERMAN was designed to handle real-world scenarios where multiple signal types are collected at uneven intervals, such as medical records with heterogeneous blood tests or event logs in complex systems. The framework combined strong predictive performance with built-in interpretability, offering importance scores at the node, graph, and subset levels. SUPERMAN allows practitioners to integrate domain priors when available, trading fine-grained interpretability for greater expressivity. Across experiments on real-world high-stakes tasks, SUPERMAN achieved state-of-the-art results. Beyond predictive accuracy, SUPERMAN's interpretability capabilities proved particularly valuable in domains like healthcare, where uncovering phase transitions and providing actionable insights is critical.

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

## A    APPENDIX

## B    THEORETICAL FRAMEWORK

### B.1    PROOF OF THEOREM 3.1

We will prove that SUPERMAN is strictly more expressive than GNAN. To prove this, we use a ground truth function that is a feature-level XOR. Let a single-node graph be endowed with binary features $x = (x_1, x_2) \in \{0, 1\}^2$ and define the target $f_\oplus(x) = x_1 \oplus x_2$.

First we will show that GNAN cannot express $f_\oplus$. A GNAN scores the graph by $\hat{y} = \sigma(\phi_1(x_1) + \phi_2(x_2))$, where each $\phi_i$ is univariate. Put $a = \phi_1(0)$, $b = \phi_1(1)$, $c = \phi_2(0)$, $d = \phi_2(1)$. To match the XOR truth-table there must exist a threshold $\tau$ such that

$$a + c < \tau, \quad b + c > \tau, \quad a + d > \tau, \quad b + d < \tau.$$

Summing the first and last inequalities yields $a + b + c + d < 2\tau$, while the middle pair gives $a + b + c + d > 2\tau$—a contradiction. Thus no GNAN realises $f_\oplus$.

Now we will show that SUPERMAN can express $f_\oplus$. Place the two features in the same subset $F = \{x_1, x_2\}$ and choose the subset-network

$$\psi_F(x_1, x_2) = x_1 + x_2 - 2x_1 x_2.$$

For the four binary inputs this mapping returns $(0, 1, 1, 0)$, exactly $f_\oplus$. Hence SUPERMAN represents a function unattainable by GNAN, proving that SUPERMAN is strictly more expressive.

### B.2    PROOF OF THEOREM 3.2

Let $S$ be a set of graphs $\{G_i\}_{j=1}^m$. Let $S_1 = \{S_i\}_{i=1}^m$ be a partition of $S$ such that $|S_i| = 1$. Let $S_2 = \{S_i\}_{i=1}^k$ such that there exists $k$ with $|S_k| > 1$. with a subset partition $\{S_i\}_{i=1}^k$. We will prove that a SUPERMAN trained over $S_2$ is strictly more expressive than a SUPERMAN trained over $S_1$.

To prove this, we use a ground truth function that is a set-level XOR. Let every graph $G_i$ carry a single binary feature $x_i \in \{0, 1\}$ and let the ExtGNAN encoder return this feature unchanged, i.e. $h(G_i) = x_i$. Denote a set containing two graphs by $S = \{G_1, G_2\}$ and define the permutation-invariant target

$$f_\oplus(S) = x_1 \oplus x_2.$$

**Singleton partition** ($S_1$).    If each graph is placed in its own subset, SUPERMAN aggregates *additively*: the model output is

$$\hat{y} = \phi(x_1) + \phi(x_2),$$

because the final SUPERMAN stage simply sums subset scores :contentReference[oaicite:0]index=0:contentReference[oaicite:1]index=1. Write $a = \phi(0)$ and $b = \phi(1)$. To realise $f_\oplus$ via a threshold $\tau$ we would need

$$a + a < \tau, \quad b + a > \tau, \quad a + b > \tau, \quad b + b < \tau.$$

Adding the first and last inequalities yields $a + b < \tau$, while the middle pair gives $a + b > \tau$—a contradiction. Hence $\text{SUPERMAN}_{S_1}$ cannot represent $f_\oplus$.

**Paired partition** ($S_2$).    Group the two graphs together and use a DeepSet $\Phi(S_2) = g(\sum_{i=1}^2 f(x_i))$ with $f(x) = x$ and $g(s) = s(2 - s)$. Then

$$g(x_1 + x_2) = \begin{cases} 0 & (x_1, x_2) = (0, 0) \text{ or } (1, 1), \\ 1 & (x_1, x_2) = (0, 1) \text{ or } (1, 0), \end{cases}$$

exactly $f_\oplus$. The final SUPERMAN sum over feature channels leaves this value unchanged, so $\text{SUPERMAN}_{S_2}$ realises $f_\oplus$.

**Strict separation.** Because $f_\oplus$ is representable by $\text{SUPERMAN}_{S_2}$ but not by $\text{SUPERMAN}_{S_1}$, the former is strictly more expressive.

### B.3 FOUR-POINT CONDITION AND RECOVERABILITY

We now turn to structural identifiability. We prove that when input graphs are connected, acyclic, and positively weighted (i.e., trees), the pairwise distance matrix learned by SUPERMAN encodes the full structure of the graph, up to isomorphism. This provides theoretical justification for the model's ability to reason over temporal structure without needing explicit graph supervision.

The following theorem shows that if the graph satisfies the four-point condition (Buneman, 1974), SUPERMAN can reconstruct the original graph from the transformed distance matrix that is fed to SUPERMAN as the graph input:

**Theorem B.1.** *Let $G$ be a graph represented by an adjacency matrix $A$, and $D$ be the transformed distance-matrix for* SUPERMAN. *Then if $D$ satisfies the four-point condition,* SUPERMAN *can learn $\rho$ such that $\rho(D) = A$.*

*Proof.* Let $G = (V, E, w)$ be a positively weighted path graph, i.e.

$$V = \{v_1, \ldots, v_n\}, \qquad E = \big\{\{v_i, v_{i+1}\} \,\big|\, i = 1, \ldots, n-1\big\}, \qquad w(\{v_i, v_{i+1}\}) > 0.$$

Define the pair-wise distance matrix $D \in \mathbb{R}^{n \times n}$ by

$$D_{uv} = \sum_{e \in P_G(u,v)} w(e) \,,$$

where $P_G(u, v)$ is the unique $u$–$v$ path in $G$. Then:

(a) Tree-metric property. $D$ satisfies the four-point condition of Buneman (Buneman, 1974); hence $(V, D)$ is a *tree metric*.

(b) Uniqueness (*no information loss*). By Buneman's theorem the tree that realises $D$ is unique up to isomorphism. For a path graph the only automorphism is the reversal $(v_1, \ldots, v_n) \mapsto (v_n, \ldots, v_1)$, so $D$ determines $G$ completely except for left–right orientation.

(c) Efficient reconstruction. $G$ can be reconstructed from $D$ in $O(n^2)$ time:

    i. Choose an endpoint $s = \arg\max_{v \in V} \max_{u \in V} D_{vu}$.
    ii. Order the vertices $v_1 = s$, $v_2, \ldots, v_n$ so that $D_{sv_1} < D_{sv_2} < \cdots < D_{sv_n}$.
    iii. Set edge weights $w(\{v_i, v_{i+1}\}) = D_{sv_{i+1}} - D_{sv_i}$ for $i = 1, \ldots, n-1$.

$\square$

## C COMPUTATIONAL COMPLEXITY AND EFFICIENT IMPLEMENTATION

In this section, we provide a big-O analysis of the time complexity of SUPERMAN. We also provide an efficient approach to implement SUPERMAN.

**Computational Complexity** The computational complexity of SUPERMAN is as follows:

- **Scales linearly with the number of graphs $m$:** Each graph is processed independently or in small subsets, so total cost is $\mathcal{O}(m)$ assuming fixed per-graph cost.
- **Scales quadratically with the number of nodes per graph $n$:** Due to the dense aggregation over all node pairs in ExtGNAN, the per-graph cost is $\mathcal{O}(n^2)$.
- **Overall complexity** for a set of graphs is:

$$\mathcal{O}\big(m \cdot K \cdot n^2 \cdot d_\psi\big)$$

where $K$ is the number of feature groups and $d_\psi$ is the cost of evaluating the multivariate neural networks.

**Adapting the complexity**  We note that while the per-graph complexity of ExtGNAN is $O(n^2)$ in the most general case, the distance function $\Delta$ can be masked to enforce any desired sparsity pattern between nodes, and therefore adapt to any complexity limitation, including a linear one. For example, one can define the $\Delta$ to only be the distance between adjacent nodes in a trajectory, and then it will be both linear in memory and time.

**Efficient Implementations**  In the main paper, we present SUPERMAN with the objective of maximal clarity, e.g., by presenting vector entry-wise operations. Nonetheless, the operations of SUPERMAN can be done in an optimized fashion for GPU, through tensor operations.

## D  DATASET DETAILS

### D.1  PHYSIONET P12

We provide the full list of the 36 physiological signals and 3 static patient features used in our experiments.

1. Alkaline phosphatase (ALP): A liver- and bone-derived enzyme; elevations suggest cholestasis, bone disease, or hepatic injury.

2. Alanine transaminase (ALT): Hepatocellular enzyme; increased values mark acute or chronic liver cell damage.

3. Aspartate transaminase (AST): Enzyme in liver, heart, and muscle; rises indicate hepatocellular or muscular injury.

4. Albumin: Major plasma protein maintaining oncotic pressure and transport; low levels reflect inflammation, malnutrition, or liver dysfunction.

5. Blood urea nitrogen (BUN): End-product of protein catabolism cleared by the kidneys; elevation signals renal impairment or high catabolic state.

6. Bilirubin: Hemoglobin breakdown product processed by the liver; accumulation indicates hepatobiliary disease or hemolysis.

7. Cholesterol: Circulating lipid essential for membranes and hormones; dysregulation is linked to cardiovascular risk.

8. Creatinine: Waste from muscle metabolism filtered by the kidneys; higher levels imply reduced glomerular filtration.

9. Invasive diastolic arterial blood pressure (DiasABP): Pressure during ventricular relaxation; low readings may reflect vasodilation or hypovolemia.

10. Fraction of inspired oxygen ($FiO_2$): Proportion of oxygen delivered; values above ambient air denote supplemental therapy.

11. Glasgow Coma Score (GCS): Composite neurologic score for eye, verbal, and motor responses; scores $\leq 8$ indicate severe impairment.

12. Glucose: Principal blood sugar; hypo- or hyper-glycemia can cause neurologic compromise and metabolic instability.

13. Serum bicarbonate ($HCO_3$): Key extracellular buffer; low levels signal metabolic acidosis, high levels metabolic alkalosis or compensation.

14. Hematocrit (HCT): Percentage of blood volume occupied by red cells; reduced values denote anemia, elevated values hemoconcentration.

15. Heart rate (HR): Beats per minute reflecting cardiac demand; tachycardia indicates stress or shock, bradycardia conduction disorders.

16. Serum potassium (K): Crucial intracellular cation; deviations predispose to dangerous arrhythmias.

17. Lactate: By-product of anaerobic metabolism; elevation marks tissue hypoxia and shock severity.

18. Invasive mean arterial blood pressure (MAP): Time-weighted average arterial pressure; low values threaten organ perfusion.

19. Mechanical ventilation flag (MechVent): Binary indicator of ventilatory support; presence denotes respiratory failure or peri-operative care.

20. Serum magnesium (Mg): Cofactor for numerous enzymatic reactions; abnormalities contribute to arrhythmias and neuromuscular instability.

21. Non-invasive diastolic arterial blood pressure (NIDiasABP): Cuff-derived diastolic pressure; trends mirror vascular tone without an arterial line.

22. Non-invasive mean arterial blood pressure (NIMAP): Cuff-based mean pressure; used when invasive monitoring is unavailable.

23. Non-invasive systolic arterial blood pressure (NISysABP): Cuff-derived systolic pressure; elevations suggest hypertension or pain response.

24. Serum sodium (Na): Principal extracellular cation governing osmolality; dysnatremias cause neurologic symptoms and fluid shifts.

25. Partial pressure of arterial carbon dioxide ($PaCO_2$): Indicator of ventilatory status; hypercapnia implies hypoventilation, hypocapnia hyperventilation.

26. Partial pressure of arterial oxygen ($PaO_2$): Measure of oxygenation efficiency; low values denote hypoxemia.

27. Arterial pH: Measure of hydrogen-ion concentration; deviations from normal reflect systemic acid–base disorders.

28. Platelet count (Platelets): Thrombocyte concentration essential for hemostasis; low counts increase bleeding risk, high counts thrombosis risk.

29. Respiration rate (RespRate): Breaths per minute; tachypnea signals metabolic acidosis or hypoxia, bradypnea central depression.

30. Hemoglobin oxygen saturation ($SaO_2$): Percentage of hemoglobin bound to oxygen; values below normal indicate significant hypoxemia.

31. Invasive systolic arterial blood pressure (SysABP): Peak pressure during ventricular ejection; extremes compromise end-organ perfusion.

32. Body temperature: Core temperature; fever suggests infection, hypothermia exposure or metabolic dysfunction.

33. Troponin I: Cardiac-specific regulatory protein; elevation confirms myocardial injury.

34. Troponin T: Isoform of cardiac troponin complex; rise parallels Troponin I in detecting myocardial necrosis.

35. Urine: Hourly urine volume as a gauge of renal perfusion; oliguria signals kidney hypoperfusion or failure.

36. White blood cell count (WBC): Reflects immune activity; leukocytosis suggests infection or stress, leukopenia marrow suppression or severe sepsis.

**Static patient features:** Age; Gender; *ICUType* – categorical code for the admitting intensive care unit (1 = Coronary Care, 2 = Cardiac Surgery Recovery, 3 = Medical ICU, 4 = Surgical ICU), capturing differences in case mix and treatment environment.

### D.2 Crohn's Disease Prediction

We detail the full list of the 17 biomarkers extracted from the Danish health registries.

1. C-reactive protein (CRP): A protein produced by the liver in response to inflammation. Elevated CRP indicates active inflammation, often associated with inflammatory diseases like CD.

2. Faecal Calprotectin (F-Cal): A protein released from neutrophils into the intestinal lumen, detectable in stool samples. Elevated levels indicate gastrointestinal inflammation and are commonly used to detect and monitor inflammatory bowel disease.

3. Leukocytes (White Blood Cells): Cells that are central to the body's immune response. Elevated leukocyte counts typically suggest infection or inflammation, including flare-ups in CD.

4. Neutrophils: A type of leukocyte involved, among other things, in fighting bacterial infections. High neutrophil counts often indicate acute inflammation or infection, including intestinal inflammation in CD.

5. Lymphocytes: A group of white blood cells that form the core of the adaptive immune system, including T cells, B cells, and natural killer (NK) cells. They are responsible for antigen-specific immune responses. Abnormal levels can signal immune dysregulation, often implicated in autoimmune and chronic inflammatory diseases such as CD.

6. Monocytes: A type of white blood cell that circulates in the blood and differentiates into macrophages or dendritic cells upon entering tissues. These cells are essential for phagocytosis, antigen presentation, and regulation of inflammation. Elevated levels may reflect immune activation or tissue damage.

7. Eosinophils: Immune cells involved primarily in allergic reactions and parasitic infections. Elevated eosinophil counts might reflect allergic responses or gastrointestinal inflammation.

8. Basophils: The least common type of leukocyte, involved in allergic and inflammatory responses. Their elevation is uncommon but may accompany certain inflammatory or allergic conditions.

9. Platelets: Cell fragments critical for blood clotting and also involved in inflammatory responses. High platelet counts (thrombocytosis) are commonly seen during active inflammation in conditions like CD.

10. Hemoglobin (Hb): The protein in red blood cells responsible for oxygen transport. Low hemoglobin (anemia) is frequently observed in chronic inflammatory conditions such as CD due to blood loss or nutrient deficiencies.

11. Iron: An essential mineral for red blood cell production. Low iron levels often indicate chronic blood loss or malabsorption, both common in CD due to intestinal inflammation.

12. Folate (Vitamin B9): A vitamin necessary for red blood cell production and DNA synthesis. Deficiency may result from impaired absorption in inflamed intestinal tissue.

13. Vitamin B12 (Cobalamin): Required for red blood cell production and neurological function. Deficiencies are common in CD, especially when the ileum is affected.

14. Vitamin D2+D3 (Ergocalciferol + Cholecalciferol): Vitamins essential for bone health and immune regulation. Low levels are often seen in CD due to malabsorption and systemic inflammation.

15. ALAT (Alanine Aminotransferase): An enzyme indicating liver function. Elevated levels may reflect liver inflammation, medication effects, or co-occurring autoimmune liver disease.

16. Albumin: A protein produced by the liver that helps maintain blood volume and transport nutrients. Low albumin can reflect chronic inflammation, malnutrition, or protein loss in CD.

17. Bilirubin: A compound produced from red blood cell breakdown. It is filtered by the liver and excreted into the intestine via bile. Elevated levels may indicate liver dysfunction, bile duct obstruction, or hemolytic anemia.

### D.2.1 CLINICAL CONTEXT AND RELATED WORK FOR PREDICTING CD ONSET

Research on predicting the onset of CD has explored a range of approaches, including the use of routinely measured blood-based biomarkers and more complex biological data derived from multi-omics technologies.

Several studies have assessed the predictive potential of standard clinical blood tests. For example, (Vestergaard et al., 2023) analyzed six routine biomarkers from 1,186 Danish patients eventually diagnosed with CD, achieving moderate predictive performance (AUROC of 0.74) approximately six months before clinical diagnosis. Larger-scale analyses, such as those using UK Biobank data, combined multiple standard biomarkers and basic demographic information, reporting similar predictive performances (AUROCs typically between 0.70–0.75). These analyses generally utilized methods like logistic regression, random forests, or gradient-boosted trees, favored for structured clinical datasets.

Other studies have integrated advanced biochemical data, known as multi-omics, including large-scale protein measurements (proteomics), metabolites (metabolomics), or genomic markers. (Garg et al., 2024) for instance, combined 67 blood biomarkers with approximately 2,900 plasma proteins from the UK Biobank, achieving an AUROC of 0.786. (Woerner et al., 2025) combined genetic risk scores with extensive proteomic data, achieving an AUROC of 0.76 for CD prediction up to five years prior to diagnosis. Similar multi-omics approaches employing microbiome profiling, immune signaling

molecules (cytokines), or lipid molecules typically achieve AUROCs between 0.75 and 0.80 but often involve significant cost, specialized laboratory analyses, and reduced consistency across diverse patient cohorts.

Overall, routine blood tests provide meaningful predictive signals for CD onset, while integrating complex biochemical measurements can improve predictive accuracy, albeit at greater cost, complexity, and variability across clinical populations.

## E   BIOMARKER SUBSET GROUPINGS

### E.1   P12

In the PhysioNet P12 task, we grouped the 36 physiological signals into one multivariate subset and 29 singleton subsets. Domain knowledge showed that only the respiratory and gas-exchange variables shared sufficiently strong, coherent dynamics to benefit from joint modeling. All other signals were physiologically diverse, so they were left as singletons to retain their unique predictive information.

1. **Arterial blood gas profile**
   *[pH, PaCO$_2$, PaO$_2$, SaO$_2$, HCO$_3$, FiO$_2$]*
   This group captures systemic acid–base status (pH, HCO$_3$), carbon dioxide clearance (PaCO$_2$), oxygenation (PaO$_2$, SaO$_2$), and inspired oxygen fraction (FiO$_2$). Together they form the canonical arterial blood gas panel, enabling the model to detect respiratory derangements such as hypoxemia, hypercapnia, or metabolic compensation.

2. **Complete blood count**
   *[WBC, HCT, Platelets]*
   This cluster summarizes hematologic composition by measuring leukocyte-mediated immune response (WBC), oxygen-carrying capacity (HCT), and clotting potential (Platelets). Joint modeling supports recognition of systemic inflammation, anemia, and coagulopathy.

3. **Comprehensive metabolic panel**
   *[Glucose, Na, K, Mg, BUN, Creatinine]*
   These biomarkers represent key substrates and electrolytes (Glucose, Na, K, Mg) and renal waste products (BUN, Creatinine). Grouping them provides a unified view of metabolic balance, electrolyte homeostasis, and kidney function.

4. **Liver function tests**
   *[ALT, AST, ALP, Albumin, Bilirubin]*
   These biomarkers assess hepatocellular injury (ALT, AST), cholestasis (ALP, Bilirubin), and hepatic synthetic function (Albumin). Their combined interpretation reflects multiple dimensions of liver health.

5. **Lipid and cardiac markers**
   *[Cholesterol, TroponinI, TroponinT, HR]*
   This group integrates lipid metabolism (Cholesterol), cardiac injury markers (Troponin I, Troponin T), and heart rate (HR). Together, they provide insight into cardiovascular stress, myocardial injury, and metabolic risk.

6. **Blood pressure profiles**
   *[SysABP, DiasABP, MAP, NISysABP, NIDiasABP, NIMAP]*
   These variables capture invasive and non-invasive arterial blood pressures, reflecting systemic hemodynamics. Grouping them enables the model to learn coherent pressure dynamics rather than treating each measurement in isolation.

7. **Ventilation mechanics**
   *[RespRate, MechVent]*
   This group reflects mechanical and physiological components of ventilation. Their joint dynamics provide context for interpreting respiratory compensation and ventilatory support.

8. **Tissue perfusion**
   *[Lactate, Urine]*
   Elevated lactate indicates anaerobic metabolism, while urine output tracks renal perfusion. Together they provide complementary signals of global tissue perfusion and shock severity.

9. **Global status indicators**
   *[GCS, Temp]*
   These variables capture overall neurologic responsiveness (GCS) and systemic temperature regulation (Temp), providing global context on patient stability and severity of illness.

## E.2 CD

In the Crohn's Disease prediction task, we grouped the 17 selected biomarkers into 7 subsets based on shared physiological function, clinical relevance, and correlated patterns observed in exploratory analyses. This configuration produced the most robust and interpretable results, balancing domain knowledge with empirical performance. The grouping is as follows:

1. **White blood cell subtypes**
   *[Leukocytes, Neutrophils, Lymphocytes, Monocytes, Eosinophils, Basophils]*
   These biomarkers all represent components of the immune system's cellular response. Grouping them enables the model to learn shared immune activation patterns, which are known to be dysregulated in inflammatory bowel diseases like CD. Combining them in a multivariate subset captures both their relative proportions and total counts, which are clinically relevant for distinguishing inflammation subtypes.

2. **Inflammation markers**
   *[CRP, Faecal Calprotectin]*
   These are key indicators of systemic and intestinal inflammation, respectively. CRP reflects acute-phase liver response, while F-Cal is specific to intestinal neutrophilic activity. Though mechanistically distinct, both are strongly correlated with inflammatory disease activity and complement each other in modeling CD-specific inflammation signatures.

3. **Platelets**
   *[Platelets]*
   Thrombocytosis (elevated platelet count) is a well-established marker of chronic inflammation. As platelet behavior is relatively independent from other hematological and nutritional markers, we model it as its own trajectory.

4. **Hemoglobin**
   *[Hemoglobin]*
   Hemoglobin concentration is a direct measure of anemia, which is prevalent in CD patients due to chronic blood loss and inflammation-induced iron sequestration. Its temporal dynamics often diverge from those of other blood components, warranting a separate representation.

5. **Iron status**
   *[Iron]*
   Iron metabolism is tightly linked to both hemoglobin levels and systemic inflammation but shows distinct dynamics. Modeling it separately allows the model to learn delayed or decoupled effects (e.g., iron deficiency preceding hemoglobin drop).

6. **Vitamin and folate status**
   *[Folate, Vitamin B12, Vitamin D2+D3]*
   These nutrients are absorbed in different regions of the gastrointestinal tract (e.g., B12 in the ileum, folate in the jejunum), and their deficiency profiles can be informative of CD location and severity. Grouping them allows the model to detect joint patterns of malabsorption and systemic nutrient depletion.

7. **Liver function markers**
   *[ALAT, Bilirubin, Albumin]*
   These biomarkers reflect hepatic function and protein synthesis. Abnormal liver enzymes and hypoalbuminemia are frequently observed in CD due to medication effects, chronic inflammation, or comorbid autoimmune liver disease. Combining them supports learning of systemic inflammatory effects beyond the gut.

This grouping reflects known biological relationships, enhances the interpretability of the model's subset-level attributions, and improves performance compared to unstructured or purely univariate representations. It enables SUPERMAN to exploit interactions among related features while maintaining a modular structure that aligns with clinical reasoning.

## F    Experimental Setup, Hyperparameter Choices and Grouping Configurations

This section outlines key implementation choices and model settings used in our experiments, including the manually tuned biomarker grouping configurations that served as an important hyperparameter for performance and interpretability.

### F.1    Hyper-Parameters

We trained all models with 100 epochs using the Adam optimizer with weight decay 1e-5. We used a ReduceLROnPlateau scheduler with a max learning rate in the 1e-2, 1e-4 range, min learning rate in the 1e-7, 1e-8 range, factor in the 0.2-0.9 range, and patience=100

We trained all models with batch size of range {16, 32}, dropout rate in {0.1, 0.2}, number of layers in the {3, 4, 5} range, hidden channels in the {32, 64} range.

Random seeds were fixed for reproducibility, and results are reported across three independent runs. All models were trained on a single NVIDIA Tesla V100-PCIE-16GB GPU.

### F.2    Grouping Configurations for Clinical Tasks

In both clinical tasks, the configuration of input feature subsets (i.e., how we grouped input biomarkers into multivariate trajectories) was treated as a manually tuned hyperparameter. These groupings determine how SUPERMAN combines individual graph representations prior to final prediction, and they affect both the expressivity and interpretability of the model.

**In-Hospital Mortality (P12).**    We compared SUPERMAN's performance under the following grouping strategies:

- **No grouping**: Each biomarker is in its own size one subset.
- **Respiratory** - one group of the biomarkers: *[FiO$_2$, PaO$_2$, PaCO$_2$, SaO$_2$, RespRate, pH, MechVent]* and the rest are singletons.
- **Metabolic Electrolytes** - one group of the biomarkers: *[Na, K, Mg, HCO3, Lactate, Glucose]* and the rest are singletons.
- **Liver Panel** - one group of the biomarkers: *[ALT, AST, ALP, Albumin, Bilirubin, Cholesterol]* and the rest are singletons.
- **Pathway-Based Grouping**: Biomarkers are organised based on their molecular or mechanistic roles, grouping them by their function in metabolism, homeostasis, or cellular composition: Energy metabolism [Glucose, Cholesterol, Lactate]; nitrogen waste clearance [BUN, Creatinine, Urine]; protein synthesis and enzymes [Albumin, ALT, AST]; liver function and cholestasis [ALP, Bilirubin]; acid–base balance [pH, HCO$_3$]; gas transport [PaO$_2$, PaCO$_2$, SaO$_2$, FiO$_2$]; mineral homeostasis [Na, K, Mg]; hematologic composition [HCT, Platelets, WBC]; cardiovascular dynamics [SysABP, DiasABP, MAP, NISysABP, NIDiasABP, NIMAP, HR]; respiratory mechanics [RespRate, MechVent]; and cardiac injury [TroponinI, TroponinT]. Global status indicators are grouped separately as [Temp, GCS].
- **Organ-System Grouping** Biomarkers are organized around organ systems and clinical monitoring domains (respiratory support, cardiovascular dynamics etc: oxygenation support [SaO$_2$, PaO$_2$, FiO$_2$, MechVent]; ventilation and acid–base balance [PaCO$_2$, RespRate, pH, HCO$_3$]; cardiovascular dynamics [HR, MAP, SysABP, DiasABP, NIMAP, NISysABP, NIDiasABP]; perfusion and renal function [Urine, Lactate, Creatinine, BUN]; hepatobiliary function [Bilirubin, ALT, AST, ALP, Albumin]; inflammation and coagulation [WBC, Platelets]; electrolyte and oxygen-carrying capacity [Na, K, Mg, HCT]; metabolic reserve [Glucose, Cholesterol]; myocardial injury [TroponinI, TroponinT]; and global status indicators [Temp, GCS].

**Crohn's Disease Onset.**    We evaluated several grouping configurations:

- **No grouping**: Each biomarker is in its own size one subset.

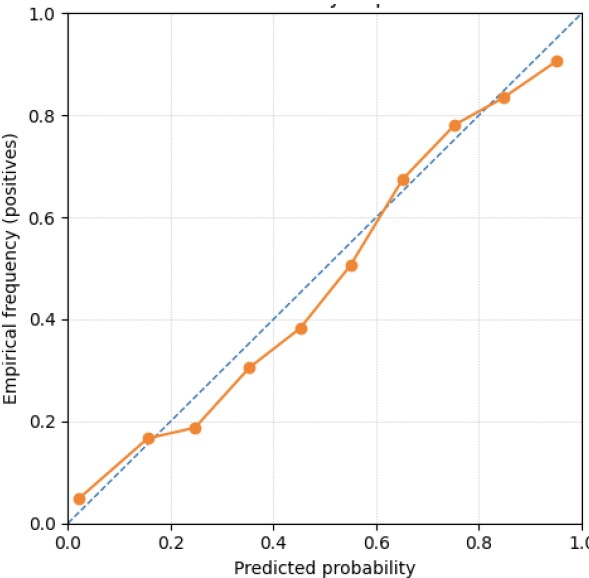

Figure 5: Q–Q calibration plot (reliability diagram) for SUPERMAN on the CD task.

- **Biologically driven grouping** (see Appendix E.2): Biomarkers are grouped into 7 clinically coherent subsets (e.g., inflammation markers, immune cell subtypes, liver function).

- **Diagnostic Panel Grouping**: A clinically motivated grouping that mirrors standard blood test panels used in routine diagnostics.

- **Data-driven grouping**: Groupings are derived from clustering biomarkers based on the complementary signal in their attribution curves (see Section 4.1.1).

- **Merged coarse groupings**: Broad categories such as inflammation, haematology, and micronutrients.

- **Minimal Pairwise Interaction** : Emphasizes minimal yet informative combinations that capture key axes of immune, inflammatory, and metabolic variational proximity.

We report the AUPRC of different biomarker groupings for predicting the onset of CD in Table 4

## G   CALIBRATION AND ROBUSTNESS ANALYSES

Here we report additional evidence on the reliability (calibration) and robustness of SUPERMAN on the Crohn's Disease (CD) task.

### G.1   CALIBRATION

We compute Expected Calibration Error (ECE) for SUPERMAN and all baselines on the CD task. ECE is reported as mean $\pm$ std over three data splits. Lower values indicate better agreement between predicted probabilities and observed outcome frequencies. As shown in Table 5, SUPERMAN attains the lowest ECE among the compared methods.

To complement ECE, we provide a Q–Q calibration plot for SUPERMANin Fig. 5. The plot compares binned predicted probabilities against empirical outcome frequencies. Close alignment with the diagonal indicates good calibration. The near-diagonal trend is consistent with SUPERMAN's low ECE.

## G.2 Noise Robustness under Controlled Distribution Shift

We evaluate robustness under controlled distribution shifts by training all models on clean data and injecting noise only at test time. For each noise level, we re-compute AUROC and AUPRC and report the relative performance drop $\delta$ (percentage change) from the clean-test baseline.

**Additive value noise.** For each biomarker trajectory graph, we perturb the primary biomarker feature (feature index $0$ corresponding to the observed biomarker value) at every node by adding zero-mean Gaussian noise with fixed standard deviation $\sigma_{\text{value}}$, independent of the feature's magnitude. If $v$ is the original biomarker value, we sample $\epsilon \sim \mathcal{N}(0, 1)$ and use $v' = v + \epsilon \cdot \sigma_{\text{value}}$. This models absolute measurement variability with a uniform noise scale. The results are shown in Table 6

**Multiplicative value noise.** Using the same feature, we instead add zero-mean Gaussian noise whose scale is proportional to the absolute feature value. If $v$ is the original value, we set the noise scale to $|v| \cdot \sigma_{\text{value}}$ and sample $v' = v + \epsilon \cdot |v| \cdot \sigma_{\text{value}} \approx v \cdot (1 + \epsilon \cdot \sigma_{\text{value}})$. This models relative measurement/units variability, where larger measurements incur larger absolute perturbations. The results are shown in Table 7

**Temporal noise.** We inject zero-mean Gaussian noise into the temporal distance matrix at test time, with standard deviation $\sigma_{\text{time}}$ (in days). This perturbs pairwise time gaps between visits while keeping biomarker values and graph structure unchanged. The results are shown in Table 8

## H LLM Usage

We relied on Large Language Models solely for grammar and spelling checks, without using them to generate or modify the scientific content.

Table 4: Performance using different biomarker subsets for CD prediction.

| Grouping Strategy | Biomarker Groups | AUPRC |
|---|---|---|
| Flat grouping | Each biomarker is modeled independently: CRP, F-Cal, Leukocytes, Neutrophils, Lymphocytes, Monocytes, Eosinophils, Basophils, Platelets, Hemoglobin, Iron, Folate, Vitamin B12, Vitamin D2+D3, ALAT, Albumin, Bilirubin. | 79.66 ± 0.96 |
| Biologically driven grouping | | 83.93 ± 0.27 |
| | • White blood cell subtypes: Leukocytes, Neutrophils, Lymphocytes, Monocytes, Eosinophils, Basophils | |
| | • Inflammatory markers: CRP, F-Cal | |
| | • Platelets | |
| | • Haemoglobin | |
| | • Iron | |
| | • Vitamin and folate status: Folate, Vitamin B12, Vitamin D2 + D3 | |
| | • Liver function markers: ALAT, Albumin, Bilirubin | |
| Diagnostic Panel Grouping | | 79.06 ± 3.4 |
| | • Inflammatory markers: CRP, F-Cal | |
| | • WBC count (main): Leukocytes, Neutrophils, Lymphocytes | |
| | • WBC rare subtypes: Monocytes, Eosinophils, Basophils | |
| | • Platelets + Hemoglobin: Platelets, Hemoglobin | |
| | • Nutrient panel: Iron, Folate, Vitamin B12, Vitamin D2+D3 | |
| | • Liver function test panel: ALAT, Albumin, Bilirubin | |
| Data-driven grouping | | 79.04 ± 7.8 |
| | • Acute Inflammation Markers: CRP, F-Cal, Platelets | |
| | • Immune–Iron Axis: Lymphocytes, Iron | |
| | • Neutrophils | |
| | • Monocytes | |
| | • Eosinophils | |
| | • Basophils | |
| | • Hemoglobin | |
| | • Folate | |
| | • Vitamin B12 | |
| | • Vitamin D | |
| | • ALAT | |
| | • Bilirubin | |
| | • Albumin | |
| | • Total Leukocytes | |
| Merged coarse groupings | | 75.21 ± 2.3 |
| | • F-Cal (local inflammation) | |
| | • Systemic immune/inflammation: CRP, Leukocytes, Neutrophils, Lymphocytes, Monocytes | |
| | • Allergy-linked eosinophils/basophils | |
| | • Hematological status: Platelets, Hemoglobin | |
| | • Nutritional status: Iron, Folate, Vitamin B12, Vitamin D2+D3 | |
| | • Hepatic status: ALAT, Albumin, Bilirubin | |
| Minimal Pairwise Interaction | | 81.50 ± 1.3 |
| | • CRP + F-Cal | |
| | • Leukocytes + Neutrophils | |
| | • Lymphocytes + Monocytes | |
| | • Eosinophils + Basophils | |
| | • Platelets | |
| | • Hemoglobin | |
| | • Iron + Folate | |
| | • Vitamin B12 + Vitamin D2+D3 | |
| | • ALAT + Albumin | |
| | • Bilirubin | |

| Model | ECE (mean $\pm$ std) |
|---|---|
| SUPERMAN | $0.028 \pm 0.004$ |
| Raindrop | $0.040 \pm 0.070$ |
| DGM2 | $0.035 \pm 0.001$ |
| mTAND | $2.16 \pm 0.54$ |
| Transformer | $3.32 \pm 1.08$ |
| Trans-mean | $3.30 \pm 1.03$ |
| SEFT | $2.40 \pm 0.71$ |

Table 5: Expected Calibration Error (ECE) on the CD task, reported as mean $\pm$ std over three splits.

| | SUPERMAN | | Raindrop | | mTAND | |
|---|---|---|---|---|---|---|
| $\sigma_v^+$ | $\Delta$AUROC | $\Delta$AUPRC | $\Delta$AUROC | $\Delta$AUPRC | $\Delta$AUROC | $\Delta$AUPRC |
| 0.0 | $0.00 \pm 0.00\%$ | $0.00 \pm 0.00\%$ | $0.00 \pm 0.00\%$ | $0.00 \pm 0.00\%$ | $0.00 \pm 0.00\%$ | $0.00 \pm 0.00\%$ |
| 0.1 | $-0.07 \pm 0.11\%$ | $-0.07 \pm 0.08\%$ | $-26.20 \pm 1.97\%$ | $-28.72 \pm 2.54\%$ | $-0.82 \pm 0.81\%$ | $-0.71 \pm 0.64\%$ |
| 0.2 | $-0.10 \pm 0.15\%$ | $-0.02 \pm 0.52\%$ | $-32.82 \pm 0.96\%$ | $-36.70 \pm 1.49\%$ | $-2.98 \pm 1.23\%$ | $-2.68 \pm 0.98\%$ |
| 0.3 | $-0.18 \pm 0.34\%$ | $-0.35 \pm 0.47\%$ | $-34.74 \pm 1.36\%$ | $-38.69 \pm 1.78\%$ | $-5.58 \pm 1.71\%$ | $-5.16 \pm 1.06\%$ |
| 0.5 | $-0.51 \pm 0.35\%$ | $-0.37 \pm 1.11\%$ | $-36.35 \pm 1.54\%$ | $-40.60 \pm 1.79\%$ | $-10.97 \pm 1.32\%$ | $-10.84 \pm 0.73\%$ |
| 0.8 | $-1.32 \pm 0.97\%$ | $-1.20 \pm 1.85\%$ | $-37.25 \pm 1.59\%$ | $-41.41 \pm 1.68\%$ | $-17.13 \pm 1.76\%$ | $-17.97 \pm 2.58\%$ |
| 1.5 | $-1.87 \pm 2.48\%$ | $-1.31 \pm 3.96\%$ | $-37.73 \pm 1.34\%$ | $-42.27 \pm 1.33\%$ | $-26.67 \pm 1.44\%$ | $-29.26 \pm 2.05\%$ |
| 3.0 | $-5.46 \pm 5.10\%$ | $-4.81 \pm 7.06\%$ | $-37.70 \pm 1.52\%$ | $-42.39 \pm 1.49\%$ | $-34.00 \pm 1.34\%$ | $-36.41 \pm 2.36\%$ |
| 5.5 | $-7.35 \pm 5.45\%$ | $-7.48 \pm 7.05\%$ | $-38.07 \pm 1.45\%$ | $-42.34 \pm 1.73\%$ | $-38.56 \pm 1.21\%$ | $-41.31 \pm 1.58\%$ |
| 7.0 | $-9.39 \pm 5.22\%$ | $-9.82 \pm 7.54\%$ | $-37.59 \pm 1.60\%$ | $-42.37 \pm 1.72\%$ | $-38.44 \pm 1.67\%$ | $-41.40 \pm 1.63\%$ |

Table 6: $\Delta$AUROC and $\Delta$AUPRC under additive value noise ($\sigma_v^+$).

| | SUPERMAN | | Raindrop | | mTAND | |
|---|---|---|---|---|---|---|
| $\sigma_v^*$ | $\Delta$AUROC | $\Delta$AUPRC | $\Delta$AUROC | $\Delta$AUPRC | $\Delta$AUROC | $\Delta$AUPRC |
| 0.0 | $0.00 \pm 0.00\%$ | $0.00 \pm 0.00\%$ | $0.00 \pm 0.00\%$ | $0.00 \pm 0.00\%$ | $0.00 \pm 0.00\%$ | $0.00 \pm 0.00\%$ |
| 0.1 | $-0.04 \pm 0.02\%$ | $0.00 \pm 0.04\%$ | $0.04 \pm 1.11\%$ | $0.02 \pm 0.65\%$ | $-0.35 \pm 0.99\%$ | $-0.35 \pm 0.63\%$ |
| 0.2 | $0.09 \pm 0.24\%$ | $0.26 \pm 0.41\%$ | $-0.21 \pm 1.15\%$ | $-0.27 \pm 0.64\%$ | $-1.36 \pm 0.75\%$ | $-1.36 \pm 0.29\%$ |
| 0.3 | $-0.15 \pm 0.48\%$ | $-0.02 \pm 0.54\%$ | $-0.40 \pm 1.20\%$ | $-0.40 \pm 0.60\%$ | $-2.98 \pm 1.21\%$ | $-3.18 \pm 1.14\%$ |
| 0.5 | $-0.57 \pm 0.56\%$ | $-0.39 \pm 0.89\%$ | $-1.28 \pm 1.31\%$ | $-1.26 \pm 0.66\%$ | $-7.61 \pm 1.40\%$ | $-7.79 \pm 1.08\%$ |
| 0.8 | $-0.78 \pm 0.66\%$ | $-0.68 \pm 1.70\%$ | $-2.45 \pm 0.99\%$ | $-2.44 \pm 0.47\%$ | $-13.79 \pm 1.30\%$ | $-13.25 \pm 1.11\%$ |
| 1.5 | $-1.82 \pm 1.98\%$ | $-1.71 \pm 3.41\%$ | $-3.19 \pm 1.57\%$ | $-3.16 \pm 1.18\%$ | $-24.22 \pm 1.96\%$ | $-22.58 \pm 1.45\%$ |
| 3.0 | $-4.63 \pm 3.22\%$ | $-4.89 \pm 5.29\%$ | $-4.56 \pm 1.18\%$ | $-4.60 \pm 0.51\%$ | $-31.53 \pm 1.61\%$ | $-29.08 \pm 1.56\%$ |
| 5.5 | $-4.91 \pm 4.85\%$ | $-5.31 \pm 5.82\%$ | $-6.41 \pm 1.51\%$ | $-6.20 \pm 0.84\%$ | $-35.16 \pm 1.09\%$ | $-32.07 \pm 1.10\%$ |
| 7.0 | $-7.45 \pm 5.28\%$ | $-8.15 \pm 6.59\%$ | $-7.23 \pm 1.80\%$ | $-7.37 \pm 0.65\%$ | $-36.74 \pm 1.82\%$ | $-33.55 \pm 2.34\%$ |

Table 7: $\Delta$AUROC and $\Delta$AUPRC under multiplicative value noise ($\sigma_v^*$).

| | SUPERMAN | | Raindrop | | mTAND | |
|---|---|---|---|---|---|---|
| $\sigma_t$ (days) | $\Delta$AUROC | $\Delta$AUPRC | $\Delta$AUROC | $\Delta$AUPRC | $\Delta$AUROC | $\Delta$AUPRC |
| 0.0 | $0.00 \pm 0.00\%$ | $0.00 \pm 0.00\%$ | $0.00 \pm 0.00\%$ | $0.00 \pm 0.00\%$ | $0.00 \pm 0.00\%$ | $0.00 \pm 0.00\%$ |
| 10.0 | $0.02 \pm 0.17\%$ | $0.03 \pm 0.16\%$ | $-0.54 \pm 1.17\%$ | $-0.72 \pm 0.44\%$ | $-2.17 \pm 0.77\%$ | $-2.16 \pm 0.65\%$ |
| 30.0 | $-0.04 \pm 0.05\%$ | $0.03 \pm 0.04\%$ | $-0.82 \pm 1.05\%$ | $-1.44 \pm 0.53\%$ | $-1.94 \pm 1.06\%$ | $-1.89 \pm 0.80\%$ |
| 90.0 | $-0.22 \pm 0.13\%$ | $-0.39 \pm 0.35\%$ | $-1.58 \pm 0.87\%$ | $-2.20 \pm 0.15\%$ | $-1.99 \pm 0.71\%$ | $-2.02 \pm 0.56\%$ |
| 150.0 | $-0.22 \pm 0.09\%$ | $-0.32 \pm 0.08\%$ | $-1.89 \pm 0.86\%$ | $-2.75 \pm 0.63\%$ | $-2.24 \pm 1.13\%$ | $-2.28 \pm 0.71\%$ |
| 300.0 | $-0.45 \pm 0.42\%$ | $-0.87 \pm 0.78\%$ | $-2.41 \pm 1.21\%$ | $-3.26 \pm 0.47\%$ | $-1.85 \pm 0.99\%$ | $-1.81 \pm 0.74\%$ |
| 500.0 | $-0.74 \pm 0.04\%$ | $-1.12 \pm 0.29\%$ | $-2.63 \pm 1.24\%$ | $-3.73 \pm 0.90\%$ | $-2.14 \pm 0.97\%$ | $-2.11 \pm 0.70\%$ |

Table 8: Relative change ($\Delta$) in AUROC and AUPRC under temporal noise.

