# OpenReview forum: "SuperMAN: Interpretable and Expressive Networks over Temporally Sparse Heterogeneous Data"
_ICLR.cc/2026/Conference — ICLR 2026 Poster_

### Official Review · Reviewer_B9RD · 2025-10-22

**Soundness:** 3
**Presentation:** 2
**Contribution:** 2
**Rating:** 4
**Confidence:** 4

**Summary:**

This paper presents GMAN (Graph additive mixing network) for learning on graphs that represent irregularly sampled time signals. Results show good empirical performance on different datasets.

**Strengths:**

- GMAN provides good empirical predictive performance and interpretability
- Domain prior can be incorporated into method
- Some theoretical analysis is included

**Weaknesses:**

- In the table 1 for example, GMAN is shown to beat other baselines, but the authors state that signals are grouped based on common biomarker domain knowledge, and tuning biomarker sets are also performed. My concern is that the benefit of GMAN is incremental in accuracy, and other methods do not use such specific domain knowledge, thus are more widely applicable and performant.
- Novelty is limited, and contribution seems incremental. Prior methods Raindrop uses GNNs and SeFT uses deep sets, and heavily relies on GNAN which also uses node level and graph level representations.
- Theoretical analysis is not extensive, seems elementary, and does not capture major elements of learning with temporal irregular signals

**Questions:**

See weaknesses.

---

> ### Author Response · Authors · 2025-11-21
>
> We thank the reviewer for the thoughtful comments.
>
> **Weakness 1** The reviewer raised a concern regarding the reliance of the SOTA performance of GMAN on specific domain knowledge. We thank the reviewer for this comment.
> We wish to clarify that the Fake News experiment does not use any domain or prior knowledge, and reaches SOTA performance. Similarly, the clinically-guided groups we examined in CD and P12 are not based on highly specialized or narrowly defined information but rather on very broad domain knowledge obtained from publicly accessible information on the internet, and we showed that this suffices to achieve SOTA performance.
>
> **Weakness 2** The reviewer claims that the novelty is limited. We thank the reviewer for this comment. To the best of our knowledge, GMAN is the first interpretable model capable of handling this type of data, while also achieving state-of-the-art performance. We therefore respectfully disagree with the claim regarding limited novelty.
> While GMAN leverages GNAN as one of its building blocks, it first extends GNAN to ExtGNAN to enable nonlinear combinations of feature groups, substantially increasing expressivity compared to the original GNAN formulation. Moreover, GNAN was designed only for single graphs, and therefore, the design for sets of graphs, to combine information across them, and to support nonlinear feature interactions, was not introduced before. This is while preserving the maximal level of interpretability within the expressivity chosen. These components represent core innovations that underpin GMAN’s contributions.
>
>
> **Weakness 3**  The reviewer claims that the theoretical analysis is not extensive. We thank the reviewer for this comment. GMAN achieves state-of-the-art performance while also being interpretable, and in this work, we provide an extensive evaluation of both its predictive performance and its interpretability properties, demonstrating how GMAN yields actionable insights in high-stakes applications. We believe this constitutes a substantial contribution in its own right.
> On the theoretical side, we provide two guarantees in Theorems. 3.1 and 3.2, establishing that combining groups of graphs and feature subsets indeed increases expressivity. These results directly validate the core design principles behind GMAN. Nonetheless, we emphasize that the primary focus of this work is not a full theoretical treatment of learning with temporally irregular signals, but rather the development and empirical validation of an interpretable and expressive model suited for such settings.

---

> ### Comment · Reviewer_B9RD · 2025-11-24
> **Review Response**
>
> I thank the authors' for their rebuttal. While I appreciate the clarifications, my concerns remain around the paper being incremental (GMAN mainly extends existing additive frameworks to sets of graphs with subset grouping). The gains in accuracy (~0.5 AUPRC) are small. In terms of intepretability, competing baselines Raindrop offers interpretability insights via inter-sensor dependency graphs and and SEFT via attention mechanisms. As far as the theory is concerned, it is narrowly focused on expressivity proofs lacking a general analysis of irregular temporal learning or a new theoretical tool.
>
> On the positive side, the paper offers an interpretable architecture with additive decomposition, integration of domain priors, good experimental results, and applications.
>
> In my opinion, the paper would add value if accepted, but it’s not clearly above the acceptance bar, so I decide to keep my score.

---

> > ### Author Response · Authors · 2025-12-02
> >
> > We thank the reviewer for reading our rebuttal and commenting further . We wish to emphasise the empirical improvements we report should not be evaluated in isolation. GMAN is not merely a slightly more accurate model, its contribution lies in achieving state-of-the-art performance while being the only model that provides intrinsic, globally consistent interpretability for irregular sets of temporal graphs.
> >
> > While the reviewer notes that Raindrop “offers interpretability insights via inter-sensor dependency graphs” and SEFT “via attention mechanisms,” these mechanisms do not constitute interpretability in the sense used in the interpretability literature. Rather, they belong to the category of post-hoc explainability.
> > Interpretability relates to models that are inherently comprehensible by design, while explainability pertains to post-hoc methods that elucidate aspects of black-box models [1, 2]. These explanations often come without correctness guarantees [2, 3, 4] and may not provide a complete description of the model and its predictions, potentially failing to expose hidden pitfalls [5, 6]. However, it has been noted that local explainability methods may not consistently align with their global counterparts [7]. Moreover, local explanations may be inadequate for verifying fairness and other risks [4].
> > In contrast, GMAN is a glass-box model that is interpretable by design. Its architecture is explicitly based on an additive decomposition across signal subsets, and therefore the explanations GMAN are not post-hoc but rather a slumps into the model itself and how actually different nodes, edges, subgraphs, or features contributed additively to the final label, as we demonstrated in Figures 2 and 3.
> >
> > [1] Doshi-Velez, F., & Kim, B. (2017). Towards a rigorous science of interpretable machine learning. arXiv:1702.08608.
> >
> > [2] Rudin, C. (2019). Stop explaining black box machine learning models for high stakes decisions and use interpretable models instead. Nature Machine Intelligence, 1(5), 206-215.
> >
> > [3] Blair Bilodeau, Natasha Jaques, Pang Wei Koh, and Been Kim. Impossibility theorems for feature attribution. Proceedings of the National Academy of Sciences, 121(2):e2304406120, 2024.
> >
> > [4] Daniel Vale, Ali El-Sharif, and Muhammed Ali. Explainable artificial intelligence (xai) post-hoc explainability methods: Risks and limitations in non-discrimination law. AI and Ethics, 2(4): 815–826, 2022.
> >
> > [5] Rebecca Wexler. When a computer program keeps you in jail. The New York Times, 13:1, 2017.
> >
> > [6] Michael McGough. How bad is sacramento’s air, exactly? google results appear at odds with reality, some say. Sacramento Bee, 7, 2018.
> >
> > [7] Gabriel Laberge, Yann Batiste Pequignot, Mario Marchand, and Foutse Khomh. Tackling the xai disagreement problem with regional explanations. In International Conference on Artificial Intelligence and Statistics, pages 2017–2025. PMLR, 2024.

---

### Official Review · Reviewer_TSAx · 2025-10-26

**Soundness:** 3
**Presentation:** 3
**Contribution:** 3
**Rating:** 6
**Confidence:** 4

**Summary:**

GMAN extends Graph Neural Additive Networks by operating on sets of graphs representing irregular, heterogeneous temporal signals.

**Strengths:**

1. Addresses an under-explored irregular-sampling problem.
2. Good design combining additive interpretability with graph-set flexibility.
3. Compelling real-world demonstrations (Crohn’s disease, LOS prediction).

**Weaknesses:**

1. Evaluation section briefly described; lacks detailed baselines beyond GNAN.
2. Interpretability examples are qualitative; quantitative faithfulness metrics missing.
3. Computational efficiency compared with standard GNNs unclear.

**Questions:**

1. For the "interpretability-expressivity trade-off," the biological grouping of biomarkers shows the best performance, but it’s unclear if data-driven grouping (e.g., clustering) can achieve similar results without domain priors. Please add experiments to validate this and discuss when domain knowledge is indispensable.
2. GMAN is applied to path graphs (medical data) and tree graphs (fake news), but it’s unclear how it performs on cyclic or disconnected graph, please add experiments on such graph types to verify generalizability beyond the tested structures.
3. The paper uses AUPRC for imbalanced medical tasks but doesn’t discuss how class imbalance is handled during training (e.g., sampling strategies, loss functions), please clarify this and provide additional metrics (e.g., AUROC) for comprehensive evaluation.
4. The ablation replaces ExtGNAN with MLPs/identity mappings, but it doesn’t test if simpler additive GNN variants (e.g., GNAN with minor modifications) can achieve comparable performance, please add such baselines to highlight GMAN’s unique value.

---

> ### Author Response · Authors · 2025-11-21
>
> We thank the reviewer for the positive assessment and helpful suggestions. We address each question in turn.
>
> **Weakness 1**  The reviewer mentioned that the evaluation section lacks detailed baselines beyond GNAN. We thank the reviewer for the input. We provided references to all the evaluated baselines in the main paper due to space limitations, and we deferred other information to the appendix.
> Following the reviewer's suggestion and with the additional page allowed in the updated version, we have addressed this by including more information on the baselines in the main body of the paper in the updated version.
>
> **Weakness 2** The reviewer commented that quantitative faithfulness metrics are missing on the Interpretability examples. We thank the reviewer for the comment.
> We would first like to emphasize that, in GMAN, interpretability is not post-hoc but built into the model architecture by design. Specifically, the contributions of nodes, graphs, or subsets are explicitly and additively used in computing the final prediction (as shown in Equations 1–4). This means that importance scores correspond directly to the actual values that are summed to produce the output label, making them faithful by construction. Unlike many post-hoc attribution methods, which estimate influence externally, GMAN’s interpretability faithfully reflects the model’s decision process by definition.
> In Section 4.1.1 and Figure 3, we go further and perform structured perturbation experiments as an empirical faithfulness proxy: by injecting noise along the first PCA component of biomarker subsets and measuring the resulting prediction change, we quantify how sensitive the model is to each group. Because GMAN is additive and interpretable by design, these changes directly trace how the model reacts to systematic perturbations in the additive terms.
> We have updated the submission in section 4.1.1 to clarify these points.
>
> **Weakness 3**  The reviewer notes that the computational efficiency of GMAN compared with standard GNNs is unclear. We thank the reviewer for the comment. We wish to point to the “COMPUTATIONAL COMPLEXITY AND EFFICIENT IMPLEMENTATION”  analysis in Appendix C.
>
>
>
> **Question 1**
> The reviewer suggests adding experiments on data-driven grouping approaches.
> We thank the reviewer for raising this point. In addition to the clinically motivated configurations, we also included a data-driven grouping strategy on the CD dataset (Table 4), where biomarkers were grouped based on similarity patterns in their attribution profiles rather than external biological knowledge. This approach requires no domain priors and allows a grouping structure to emerge directly from the data.
> Nonetheless, we note that the  Fake News experiments do not use any domain or prior knowledge and reach SOTA performance, and the clinically guided groups used in CD and P12 are not based on highly specialized or narrow information, but rather on broad domain knowledge, and we showed that this suffices to achieve SOTA performance.
> We have updated the submission to clarify this point in Section 4.1.
>
>
> **Question 2**:
> The reviewer asks about the performance of GMAN on cyclic or disconnected graphs.
> We thank the reviewer for this question. We first note that GMAN can be applied to arbitrary graphs, including directed, undirected, connected or disconnected.
> Specifically in the fake-news experiment, the sets of graphs of single nodes are used as one disconnected graph as detailed in Section 4.2 and shown in Figure 4. As shown, GMAN provides SOTA performance on this dataset.
>
>
>
> **Question 3**:
> The reviewer suggests adding more information on how imbalanceness in the data is handled, and other than the commonly used one for the CD and P12 tasks, which is PRAUC. We thank the reviewer for the comment. For both datasets, we performed downsampling of the data to balance it. We have updated the submission to include this information in Section 4.1.
> Following the reviewer's suggestion, we also repeated the experiments with AUROC. Unfortunately, our compute is very limited and therefore we are not able to provide these yet. We will share these results as soon as pos,sible within the rebuttal period once the runs are done.
>
> **Question 4**:
>
> The reviewer suggests that, in the ablation studies replacing ExtGNAN, we should also examine simpler additive GNN variants in addition to MLP/identity, specifically recommending examining GNAN as an alternative to ExtGNAN.
> We thank the reviewer for the suggestion. We would like to point the reviewer to row (v) in Table 3 of the ablation studies, where we do exactly this. In this ablation, ExtGNAN is replaced with GNAN while keeping the rest of the GMAN architecture unchanged. The performance of this variant is substantially lower (−4.38% ± 2.85 AUPRC), indicating that ExtGNAN contributes meaningful capability beyond GNAN.

---

> > ### Comment · Reviewer_TSAx · 2025-11-23
> >
> > Thanks for the clarifications and added details. Could you upload a version with changes highlighted?
> >
> > After reading the revised Appendix, I have one suggestion. Some claims in the manuscript are stated too strongly, but the limitations that should be mentioned:
> >
> > 1. Computational feasibility: GMAN aggregates over all node pairs in a graph, which gives O(m k n^2 d) complexity. This makes it hard to apply to large networks (thousands of nodes).
> >
> > 2. Method scope: GMAN is a conceptually nice, interpretable model for irregular multi-signal temporal data. It is not necessarily more powerful than standard GNNs, and it does not have a more efficient structure. The grouping is a manually designed inductive bias.
> >
> > I am keeping my current score.

---

> > > ### Author Response · Authors · 2025-11-23
> > >
> > > We thank the reviewer for reading our rebuttal, for the additional the thoughtful suggestions and for maintaining the high score. We updated the revision with boldface on all the edits we added in the main paper and in the Appendix.
> > >
> > > **1.** We wish to note that while the per-graph complexity of ExtGNAN is $O(n^2)$ in the most general case, the distance function $\Delta$ can be masked to enforce any desired sparsity pattern between nodes, and therefore adapt to any complexity limitation, including a linear one.
> > > For example, one can define the $\Delta$ to only be the distance between adjacent nodes in a trajectory, and then it will be both linear in memory and time, to easily adapt for large-scale graphs. This is similar to the squared attention complexity in transformers, that can be masked to reduce the complexity overhead, if needed.
> > > We added this clarification in Appendix C (in boldface).
> > >
> > > **2.** Regarding the scope of the method, we agree with the reviewer’s suggestions. However, we note that we do not claim anywhere in the paper that GMAN 'has a more efficient structure' or is 'more powerful than standard GNNs' in an absolute sense. Rather, our empirical results show that it outperforms several commonly used GNN baselines and even transformers, while also providing multiple levels of interpretability, which is a crucial property in many high-stakes applications. We therefore argue that GMAN can offer substantial practical value.
> > >
> > > We are happy to address any further concern or questions.

---

> > > > ### Comment · Reviewer_TSAx · 2025-11-24
> > > >
> > > > I’d suggest that the authors avoid using terms like “novel” in the abstract, and also try not to repeat "SOTA" too many times. As written, these phrases make the paper sound like it’s claiming to be the best method for all real-world high-stakes tasks, which isn’t really supported. You only evaluate on a few datasets. It would read much better if the claims were toned down a bit and kept closer to what the experiments actually show.

---

> > > > > ### Author Response · Authors · 2025-11-24
> > > > >
> > > > > We thank the reviewer for the kind suggestions. We appreciate the valuable input and will take it into consideration.

---

### Official Review · Reviewer_PX2M · 2025-10-29

**Soundness:** 3
**Presentation:** 3
**Contribution:** 3
**Rating:** 6
**Confidence:** 4

**Summary:**

This paper proposes GMAN, a model for learning from a set of graphs that represent irregular, asynchronous signals (e.g., per-biomarker paths in EHR; propagation trees in social media). The key idea is to (i) encode each graph with an Extended GNAN (ExtGNAN) that allows grouped (multivariate) feature functions to trade fine-grained feature-level interpretability for higher expressivity, and then (ii) mix multiple graphs via a DeepSets aggregator at the subset level, yielding additive scores at the node/graph/subset levels and thus transparent importance attributions. The paper proves GMAN is strictly more expressive than GNAN and that grouping multiple graphs into subsets further strictly increases expressivity; it also gives a recoverability result (four-point condition) and a complexity analysis. Empirically, GMAN is evaluated on ICU length-of-stay (PhysioNet 2012), Crohn’s disease onset, and GossipCop fake-news detection, reporting competitive performance with built-in interpretability; ablations show non-trivial drops when removing the DeepSets mixer, the distance kernel ρ, or replacing ExtGNAN.

**Strengths:**

* Well-motivated representation of irregular, multi-signal data: Modeling each signal trajectory as a path graph and learning over a set of such graphs avoids time-gridding/imputation artifacts; the paper also justifies recoverability of structure under a tree-metric condition and gives a clear O($m·K·n²·d\psi$) complexity with GPU-friendly implementation notes.
* Theory for interpretability–expressivity trade-off: ExtGNAN enables multivariate within-graph feature grouping; the authors prove GMAN > GNAN in expressivity and that subset-mixing of multiple graphs (DeepSets) is strictly more expressive than singleton mixing, aligning with additive-model literature and permutation-invariant set learning.
* Built-in, multi-grain interpretability that surfaces plausible clinical signals: Node-, graph-, and subset-level importances are direct contributions (not post-hoc), and qualitative analyses on P12/CD highlight clinically coherent markers (e.g., F-Cal/CRP; renal and hepatic panels) and meaningful subset curves.
* Ablations substantiate design choices: Replacing DeepSets with mean pooling, removing ρ, or downgrading ExtGNAN causes notable AUPRC drops (e.g., −20% with mean pooling; −12% without ρ), supporting the necessity of the proposed components.
* Experiments span clinical time-series and social-media propagation graphs (GossipCop) with appropriate baselines per structure family (sequence-like vs. general graphs), and GMAN outperforms strong GNN baselines on GossipCop.

**Weaknesses:**

* Novelty relative to GNAN could be made crisper in the writing: While theorems show strict expressivity gains, the narrative at times reads like an incremental extension of GNAN; emphasize the two-level mixing (within-graph feature grouping and across-graph subset mixing) and the new recoverability/complexity results as core contributions.

* Scalability: ExtGNAN performs dense pairwise aggregation across nodes, giving per-graph O(n²) cost and overall O(m·K·n²·dψ); this may limit very long trajectories (ICU with minute-level vitals) unless optimized kernels or sparsification are used. Benchmarks on long sequences would strengthen the case.

* Limited evidence of robustness & calibration: Results focus on discrimination (e.g., AUPRC/accuracy). Reliability (calibration curves/ECE), OOD generalization (e.g., cross-hospital or temporal drift), and fairness stratifications are missing; these are common asks in clinical ML.

* Codes are not avaialble at this stage, limiting its reproducibility

**Questions:**

* Can you report calibration (ECE/Reliability) and robustness (noise stress tests, temporal shift) on P12/CD?
* How does performance/interpretability scale beyond 48-hour P12 windows—can GMAN handle much denser ICU streams without subsampling given the O(n²) component? Any sparse kernel or Nyström-like approximation?

---

> ### Author Response · Authors · 2025-11-21
>
> **Weakness 1**
> The reviewer suggests improving the writing to emphasize the novelty of GMAN relative to GNAN. We thank the reviewer for this suggestion. We have updated the submission to make this part clearer in the introduction Section. Specifically, we clarify that:
> (a) GNAN is limited to single-graph inputs, whereas GMAN is expressly designed for sets of graphs, enabling learning from collections of heterogeneous signals..
> (b) Within each graph, GMAN introduces Extended GNAN (ExtGNAN), which extends GNAN’s univariate functions with multivariate feature groups, capturing nonlinear dependencies among related features while retaining additive transparency at the group level. ExtGNAN thus serves as GMAN’s atomic building block and directly expands GNAN’s representational capacity.
> (c) Beyond this intra-graph extension, GMAN further introduces a graph-grouping mechanism, which is not available to GNAN as GNAN only acts on single graphs.
>
>
>
> **Weakness 2**  The reviewer notes that ExtGNAN has an O(n^2) per-graph cost, raising concerns about scalability to very long trajectories and requesting evidence or benchmarks demonstrating feasibility.
> We thank the reviewer for raising this important point. First, we note that while the per-graph complexity of ExtGNAN is $O(n^2)$ in the most general case, the distance function $\Delta$ can be masked to enforce any desired sparsity pattern between nodes, and therefore adapt to any complexity limitation, including a linear one. For example, one can define the $\Delta$ to only be the distance between adjacent nodes in a trajectory, and then it will be both linear in memory and time.  We added this point to Appendix C with this information.
> Nonetheless, following the reviewer comment, we conducted a synthetic experiment to examine the act on very long trajectories, and to illustrate the ability to adapt $\Delta$ to linear complexity.
> We generated a synthetic dataset with 1000 samples, each consisting of 100 trajectories, each with 1000 points. On this dataset, the dense (unmasked) version required ~48 seconds per epoch on an Nvidia GeForce RTX 3090 GPU. Applying a linear-time mask, which allowed each node to aggregate only from the node before it in the trajectory, reduced the epoch time to ~7 seconds, achieving a 7x speed-up. This demonstrates that GMAN can scale to long trajectories as is and can also be easily adapted to address computational and runtime limitations through flexible masking, while preserving interpretability.
>
> **Weakness 3** The reviewer suggests adding empirical evidence on Reliability, OOD generalization, and fairness stratifications, which are common tasks in clinical ML. We thank the reviewer for the comment. While GMAN is not limited to clinical ML, following the reviewer’s comment we conducted the following experiments, and added them under Appendix G: ADDITIONAL CALIBRATION AND ROBUSTNESS ANALYSES, including figures we are not able to attach within our answer here:
>
> **1.** Expected Calibration Error (ECE). We computed ECE ( mean ± std over 3 splits) for GMAN and baselines on the CD task. As the following table shows, GMAN achieves the lowest ECE, indicating better calibration of predicted probabilities.
>
> | Model        | ECE (mean ± std) |
> |--------------|------------------|
> | GMAN         | 0.028 ± 0.004    |
> | Raindrop     | 0.040 ± 0.070    |
> | DGM2         | 0.035 ± 0.001    |
> | mTAND        | 2.16 ± 0.54      |
> | Transformer  | 3.32 ± 1.08      |
> | Trans-mean   | 3.30 ± 1.03      |
> | SEFT         | 2.40 ± 0.71      |
>
> **2.** We additionally provide a Q–Q calibration plot for GMAN, which shows the predicted probabilities closely tracking empirical outcome frequencies across bins, consistent with the low ECE ≈ 0.03 and supporting that GMAN is not only discriminative but also well calibrated. For practical reasons, we have included the plot in the revised appendix (Appendix G: ADDITIONAL CALIBRATION AND ROBUSTNESS ANALYSES, Figure 5)
>
> **3.**Finally, to probe robustness under controlled distribution shift, we ran three complementary noise experiments. In all cases, models are trained on clean data, and noise is injected only at test time while we sweep the noise level and re-compute AUPRC. We report the drop in performance (\delta) from the model’s baseline test performance.
>
> **a. Additive value noise.**
> For each graph and biomarker, we perturb the primary biomarker feature (feature index 0, corresponding to the value of the biomarker) at every node by adding zero-mean Gaussian noise with a fixed standard deviation $\sigma_{\text{value}}$, independent of the feature’s magnitude. Concretely, if $v$ is the original biomarker value, we sample $\epsilon \sim \mathcal{N}(0, 1)$ and use
>
> $$
> v' = v + \epsilon \cdot \sigma_{\text{value}}.
> $$
>
> This corresponds to an absolute perturbation of the measurement, with the same scale applied everywhere.
>
>
> (continued in next comment)

---

> > ### Author Response · Authors · 2025-11-21
> >
> > **ΔAUROC / ΔAUPRC under additive value noise (σᵥ⁺)**
> >
> > | σᵥ⁺ | **GMAN ΔAUROC** | **GMAN ΔAUPRC** | **Raindrop ΔAUROC** | **Raindrop ΔAUPRC** | **mTAND ΔAUROC** | **mTAND ΔAUPRC** |
> > |-----|------------------|------------------|----------------------|----------------------|-------------------|-------------------|
> > | 0.0 | 0.00 ± 0.00% | 0.00 ± 0.00% | 0.00 ± 0.00% | 0.00 ± 0.00% | 0.00 ± 0.00% | 0.00 ± 0.00% |
> > | 0.1 | -0.07 ± 0.11% | -0.07 ± 0.08% | -26.20 ± 1.97% | -28.72 ± 2.54% | -0.82 ± 0.81% | -0.71 ± 0.64% |
> > | 0.2 | -0.10 ± 0.15% | -0.02 ± 0.52% | -32.82 ± 0.96% | -36.70 ± 1.49% | -2.98 ± 1.23% | -2.68 ± 0.98% |
> > | 0.3 | -0.18 ± 0.34% | -0.35 ± 0.47% | -34.74 ± 1.36% | -38.69 ± 1.78% | -5.58 ± 1.71% | -5.16 ± 1.06% |
> > | 0.5 | -0.51 ± 0.35% | -0.37 ± 1.11% | -36.35 ± 1.54% | -40.60 ± 1.79% | -10.97 ± 1.32% | -10.84 ± 0.73% |
> > | 0.8 | -1.32 ± 0.97% | -1.20 ± 1.85% | -37.25 ± 1.59% | -41.41 ± 1.68% | -17.13 ± 1.76% | -17.97 ± 2.58% |
> > | 1.5 | -1.87 ± 2.48% | -1.31 ± 3.96% | -37.73 ± 1.34% | -42.27 ± 1.33% | -26.67 ± 1.44% | -29.26 ± 2.05% |
> > | 3.0 | -5.46 ± 5.10% | -4.81 ± 7.06% | -37.70 ± 1.52% | -42.39 ± 1.49% | -34.00 ± 1.34% | -36.41 ± 2.36% |
> > | 5.5 | -7.35 ± 5.45% | -7.48 ± 7.05% | -38.07 ± 1.45% | -42.34 ± 1.73% | -38.56 ± 1.21% | -41.31 ± 1.58% |
> > | 7.0 | -9.39 ± 5.22% | -9.82 ± 7.54% | -37.59 ± 1.60% | -42.37 ± 1.72% | -38.44 ± 1.67% | -41.40 ± 1.63% |
> >
> >
> >
> > *b.*
> > Multiplicative value noise: Using the same biomarker feature, we instead add zero-mean Gaussian noise whose standard deviation is proportional to the absolute value of the feature. If $v$ is the original value, we set the noise scale to $|v| \cdot \sigma_{\text{value}}$ and sample
> >
> > $$
> > v' = v + \epsilon \cdot |v| \cdot \sigma_{\text{value}}
> >       \;\approx\; v \cdot (1 + \epsilon \cdot \sigma_{\text{value}}),
> > $$
> >
> > so the perturbation is approximately *relative* to the size of the biomarker (small values receive small absolute noise; large values receive larger absolute noise).
> >
> >
> > **ΔAUROC / ΔAUPRC under multiplicative value noise (σᵥ\*)**
> >
> > | σᵥ\* | **GMAN ΔAUROC** | **GMAN ΔAUPRC** | **Raindrop ΔAUROC** | **Raindrop ΔAUPRC** | **mTAND ΔAUROC** | **mTAND ΔAUPRC** |
> > |------|------------------|------------------|----------------------|----------------------|-------------------|-------------------|
> > | 0.0 | 0.00 ± 0.00% | 0.00 ± 0.00% | 0.00 ± 0.00% | 0.00 ± 0.00% | 0.00 ± 0.00% | 0.00 ± 0.00% |
> > | 0.1 | -0.04 ± 0.02% | 0.00 ± 0.04% | 0.04 ± 1.11% | 0.02 ± 0.65% | -0.35 ± 0.99% | -0.35 ± 0.63% |
> > | 0.2 | 0.09 ± 0.24% | 0.26 ± 0.41% | -0.21 ± 1.15% | -0.27 ± 0.64% | -1.36 ± 0.75% | -1.36 ± 0.29% |
> > | 0.3 | -0.15 ± 0.48% | -0.02 ± 0.54% | -0.40 ± 1.20% | -0.40 ± 0.60% | -2.98 ± 1.21% | -3.18 ± 1.14% |
> > | 0.5 | -0.57 ± 0.56% | -0.39 ± 0.89% | -1.28 ± 1.31% | -1.26 ± 0.66% | -7.61 ± 1.40% | -7.79 ± 1.08% |
> > | 0.8 | -0.78 ± 0.66% | -0.68 ± 1.70% | -2.45 ± 0.99% | -2.44 ± 0.47% | -13.79 ± 1.30% | -13.25 ± 1.11% |
> > | 1.5 | -1.82 ± 1.98% | -1.71 ± 3.41% | -3.19 ± 1.57% | -3.16 ± 1.18% | -24.22 ± 1.96% | -22.58 ± 1.45% |
> > | 3.0 | -4.63 ± 3.22% | -4.89 ± 5.29% | -4.56 ± 1.18% | -4.60 ± 0.51% | -31.53 ± 1.61% | -29.08 ± 1.56% |
> > | 5.5 | -4.91 ± 4.85% | -5.31 ± 5.82% | -6.41 ± 1.51% | -6.20 ± 0.84% | -35.16 ± 1.09% | -32.07 ± 1.10% |
> > | 7.0 | -7.45 ± 5.28% | -8.15 ± 6.59% | -7.23 ± 1.80% | -7.37 ± 0.65% | -36.74 ± 1.82% | -33.55 ± 2.34% |
> >
> > **c.** Temporal noise: we inject zero-mean Gaussian noise into the temporal distance matrix at test time, with standard deviation σ_time (in days), thereby perturbing the pairwise time gaps between visits while leaving all other inputs unchanged.
> > **ΔAUROC / ΔAUPRC under temporal noise (σₜ, days)**
> > | σₜ (days) | **GMAN ΔAUROC** | **GMAN ΔAUPRC** | **Raindrop ΔAUROC** | **Raindrop ΔAUPRC** | **mTAND ΔAUROC** | **mTAND ΔAUPRC** |
> > |-----------|------------------|------------------|----------------------|----------------------|-------------------|-------------------|
> > | 0.0   | 0.00 ± 0.00% | 0.00 ± 0.00% | 0.00 ± 0.00% | 0.00 ± 0.00% | 0.00 ± 0.00% | 0.00 ± 0.00% |
> > | 10.0  | 0.02 ± 0.17% | 0.03 ± 0.16% | -0.54 ± 1.17% | -0.72 ± 0.44% | -2.17 ± 0.77% | -2.16 ± 0.65% |
> > | 30.0  | -0.04 ± 0.05% | 0.03 ± 0.04% | -0.82 ± 1.05% | -1.44 ± 0.53% | -1.94 ± 1.06% | -1.89 ± 0.80% |
> > | 90.0  | -0.22 ± 0.13% | -0.39 ± 0.35% | -1.58 ± 0.87% | -2.20 ± 0.15% | -1.99 ± 0.71% | -2.02 ± 0.56% |
> > | 150.0 | -0.22 ± 0.09% | -0.32 ± 0.08% | -1.89 ± 0.86% | -2.75 ± 0.63% | -2.24 ± 1.13% | -2.28 ± 0.71% |
> > | 300.0 | -0.45 ± 0.42% | -0.87 ± 0.78% | -2.41 ± 1.21% | -3.26 ± 0.47% | -1.85 ± 0.99% | -1.81 ± 0.74% |
> > | 500.0 | -0.74 ± 0.04% | -1.12 ± 0.29% | -2.63 ± 1.24% | -3.73 ± 0.90% | -2.14 ± 0.97% | -2.11 ± 0.70% |

---

> > > ### Author Response · Authors · 2025-11-21
> > >
> > > **Weakness 4** The reviewer claims code was not supplied. The code with all running instructions for reproducibility was already provided in the Supplementary Material zip file.
> > >
> > > **Question 1**  The reviewer asks for calibration and robustness results. We address this in Weakness 3.
> > >
> > > **Question 2** The reviewer asks how GMAN scales for long trajectories. This is addressed in Weakness 2.

---

> ### Comment · Reviewer_PX2M · 2025-11-24
>
> I thank the authors for their rebuttal. I think the authors have performed substantial experiments and my concerns have been well-addressed. I think now that the paper has been strengthened that the advantages are more significant compared to the weaknesses, so I have increased my scores.

---

> > ### Author Response · Authors · 2025-11-24
> >
> > We thank the reviewer for carefully reading our rebuttal, for acknowledging the merits of our work, and for raising the score. We would be happy to address any further concerns.

---

### Official Review · Reviewer_efJq · 2025-11-02

**Soundness:** 2
**Presentation:** 3
**Contribution:** 2
**Rating:** 6
**Confidence:** 4

**Summary:**

This paper introduces Graph Mixing Additive Networks (GMAN), a novel, interpretable framework for learning from sets of sparse, irregular temporal graph signals. Using Extended Graph Neural Additive Networks (ExtGNAN), GMAN models nonlinear interactions within signal groups while preserving additive interpretability at the node, graph, or subset level. This approach balances expressiveness and interpretability, allowing user-defined signal grouping (domain knowledge) to enhance representational power. GMAN is theoretically proven more expressive than prior models like GNAN. It achieves state-of-the-art performance on high-stakes tasks (e.g., Crohn’s disease prediction, fake news detection), delivering meaningful insights via built-in attributions.

**Strengths:**

1. It handles irregular, heterogeneous data: GMAN introduces a novel way to learn from sets of sparse, irregular time-series signals without any resampling or imputation. By converting each signal into a graph (e.g. linking events by time), the model preserves timing gaps and irregular patterns that other methods might obscure. This design avoids the information loss incurred by aligning or filling in missing data, enabling the model to exploit the full richness of the raw data (e.g. varying intervals between events can inform the prediction).

2. Interpretable at Multiple Granularities: A key strength of GMAN is its inherently interpretable architecture. The additive structure means one can decompose the prediction into contributions from individual nodes, entire graphs, or subsets of graphs. It retains the feature-level and node-level interpretability of GNAN for those parts of the model that remain ungrouped, and additionally provides graph-level and subset-level importance scores for grouped components. These transparent contributions are directly linked to the final output (since GMAN sums them linearly), making it straightforward to explain what drove a prediction. This multi-level interpretability is especially valuable in domains like healthcare, where practitioners might ask which biomarkers (or groups of biomarkers) were most influential in a diagnosis.

3. Flexible Expressiveness via Grouping: GMAN offers a trade-off between model complexity and interpretability. By grouping features or signals, domain experts can inject prior knowledge about which inputs actually interact, enabling the model to capture nonlinear interactions within those groups. The theory guarantees that any such grouping strictly increases the model’s expressive power compared to keeping everything separate. Practically, this flexibility led to state-of-the-art performance in diverse tasks – from clinical predictions to fake news detection – indicating that GMAN can adapt to different data structures and learn complex patterns when needed. Notably, its success across domains (time-series medicine and graph-structured social data) showcases the approach’s generality and robustness. Additionally, the use of a learned time-distance function ρ in ExtGNAN means GMAN can flexibly model temporal influence (e.g. decaying or long-range effects) rather than relying on fixed time windows, which is a powerful design for temporal graphs.

**Weaknesses:**

1. While GMAN’s grouping mechanism is powerful, it requires a priori decisions about how to partition features or signals. The model’s performance can depend on choosing sensible groups – a process that may need domain expertise or extensive tuning. In the experiments, the authors manually tried several grouping schemes (including no grouping vs. clinically guided groupings) and selected the best performing one. This indicates an added complexity: users must either have prior knowledge to guide grouping or resort to hyperparameter search to find an optimal grouping strategy. In scenarios with little known domain structure, finding the right grouping could be challenging. Moreover, if signals that truly interact are mistakenly kept separate (or vice versa), the additive structure of GMAN might fail to capture those cross-signal effects, potentially hurting performance in such cases.

2. GMAN cannot maximize expressiveness and maintain full fine-grain interpretability at the same time – there is an inherent trade-off. When features are grouped or signals are combined into a subset, the model loses the ability to attribute importance to each individual feature or each individual graph within that group. Interpretability is then only available at the aggregate subset level. In practice this means, for example, one might know a collection of biomarkers as a whole was important, but not the exact contribution of each single biomarker in that group. If a use case demands insight at the most granular level for every feature/signal, GMAN would need to forego grouping – but then it essentially reduces to GNAN, which might yield weaker accuracy if important interactions exist. Thus, in scenarios where every individual feature’s impact needs to be distinctly understood, GMAN’s strength (grouped nonlinear modeling) becomes limited unless one is willing to sacrifice some clarity.

3. The GMAN architecture introduces more components than a standard model, which could raise computational complexity. Each subset of graphs has its own ExtGNAN (each with multiple neural networks for feature groups) and possibly a DeepSets aggregator, and each node in a graph aggregates messages from all other nodes (weighted by ρ). This additive all-pairs message passing can be costly for large graphs, potentially $O(n^2)$ per graph in the worst case. In their experiments, this was manageable (e.g. path graphs or moderate-sized biomedical time series), but for very large graphs or very large numbers of signals, training and inference might slow down. Additionally, representing certain data as a set of many graphs could inflate memory or computation: for instance, breaking a big tree into hundreds of path graphs means running many small GNN instances. There is also a risk of model complexity in terms of parameters – with separate neural networks for each feature group and each subset, the number of parameters grows, which might require careful regularization and ample data to avoid overfitting. While not reported as an issue in the paper, these factors suggest that GMAN could be less sample-efficient or slower to train compared to simpler architectures, especially if used naively on very large-scale problems.

**Questions:**

The proofs demonstrate that grouping signals increases expressivity (Theorems 3.1–3.2). Could the authors clarify what classes of real-world functions this additional expressivity enables beyond XOR-type examples, perhaps with a more domain-relevant illustration (e.g., nonlinear biomarker interactions)?

The model’s flexibility depends on user-defined graph/feature grouping. Could the authors discuss automatic or data-driven methods for learning optimal groupings, and whether improper grouping choices can significantly degrade performance?

---

> ### Author Response · Authors · 2025-11-21
>
> We thank the reviewer for the thoughtful, constructive assessment. We address each weakness and question in turn.
>
> **Weakness 1**:  The reviewer raised a question regarding the grouping mechanism and suggested that prior knowledge may be required to optimize GMAN. We thank the reviewer for this insightful comment.
> First, in the Fake News experiments, we did not use any domain knowledge and achieved SOTA performance. The clinically-guided groups we examined in CD and P12 are not based on highly specialized or narrowly defined information but rather on a very broad domain knowledge, and we showed that this suffices to achieve SOTA performance.
> The grouping mechanism in GMAN is an optional capability rather than a requirement. It serves as an additional degree of flexibility beyond GNAN, allowing the model to maximize expressivity when full interpretability is not essential, unlike GNAN, which enforces complete interpretability at the cost of expressivity. This flexibility allows practitioners to directly exploit their domain knowledge by grouping variables they believe to interact mechanistically, if such knowledge exists. Importantly, such priors often exist naturally.
> In this sense, grouping is not an arbitrary hyperparameter but a controlled design choice users may or may not utilize. Overall, the flexibility that GMAN offers enables users to make informed design choices that incorporate domain-specific priors while retaining the ability to default to ungrouped settings when such priors are unavailable. We have updated the submission to clarify this point in Section 4.1.
>
> **Weakness 2**:   The reviewer suggests that GMAN cannot simultaneously achieve maximal expressivity and full fine-grained interpretability. We thank the reviewer for this insightful comment. We would first like to emphasize that GMAN achieves state-of-the-art performance compared to black-box models that offer no interpretability, while also providing multiple levels of interpretability.
> Indeed, GMAN maximizes its expressivity within each chosen grouping configuration. The trade-off between interpretability and expressivity is intentionally designed to be user-controlled: when full interpretability is selected, GMAN remains maximally expressive within that chosen interpretable regime. This is while other models do not allow for ANY level of interpretability.
>
>
> **Weakness 3**: The reviewer noted that GMAN introduces more components than “standard” models and mentioned the worst case complexity of $O(n^2)$ of ExtGNAN. We thank the reviewer for this comment. However, it is not entirely clear what constitutes a “standard” model in this context, nor what level of granularity is implied by the term “component.” For example, a transformer (used as one of our baselines) includes token and positional embeddings, query, key, and value projections, an attention output projection, feed-forward layers, layer normalizations, residual connections, and an output head, totaling roughly 11 core components per layer. This is arguably more than GMAN, depending on how one defines a component. Nevertheless, GMAN empirically outperforms such models while additionally offering interpretability, which standard architectures do not provide. We also emphasize that the component breakdowns presented in the paper are intended purely for clarity.
> Regarding ExtGNAN, we note that while the per-graph complexity of ExtGNAN is $O(n^2)$ in the most general case, the distance function $\Delta$ can be masked to enforce any desired sparsity pattern between nodes, and therefore adapt to any complexity limitation, including a linear one. For example, one can define the $\Delta$ to only be the distance between adjacent nodes in a trajectory, and then it will be both linear in memory and time.  We added this point to Appendix C with this information.

---

> > ### Author Response · Authors · 2025-11-21
> >
> > **Question 1**:  The reviewer asks for real-world examples on non-linear interactions between groups, over the XOR function used to prove Theorems 3.1–3.2 .  We thank the reviewer for this insightful question.
> > While Theorems 3.1–3.2 use XOR as a minimal illustrative example, the gain in expressivity directly translates to real-world nonlinear dependencies between signals. In the medical experiments (Section 4.1) grouping enables GMAN to capture nonlinear interactions between physiologically related biomarkers. For instance, clinically established composites such as the neutrophil-to-lymphocyte ratio (NLR) reflect a non-additive immune-balance effect, and the albumin–bilirubin (ALBI) score combines albumin and bilirubin through a nonlinear, log-weighted relationship rather than independent additive contributions.
> >
> > **Question 2**: The reviewer asks about data-driven grouping approaches and the potential impact of suboptimal grouping selections. We thank the reviewer for these important questions. As elaborated in our response to Weakness 1, grouping is typically guided by existing domain knowledge but this is not a strict requirement. The concern about inappropriate group selection is analogous to that of feature selection in all models and thus a general modeling consideration. Regarding data-driven grouping, we examined this in Table 4, where we report a clustering-based approach that automatically groups biomarkers according to the similarity of their attribution curves, achieving performance comparable to manually defined schemes. We have updated the submission to make this point more explicit.

---

### Meta-Review · Area_Chair_wKXr · 2026-01-06

**Summary:**

This paper proposes GMAN, a graph-mixing neural additive model for learning from sets of irregular temporal graphs, combining subset-level grouping with built-in multigrain interpretability.

- Across the reviews, there is broad agreement that the paper addresses an important and under-explored setting, offers a coherent and interpretable modeling framework, and provides theoretical and empirical support for the proposed grouping mechanism.
- The additive structure provides transparent node / graph / subset-level interpretability, which is especially valuable in clinical applications.
- Experiments and ablations support the necessity of the proposed components, and results are competitive or strong across domains.
- Some concerns by the only negative reviewer seem to be addressed.

Therefore, I recommend the accept.

**Reviewer Concerns:**

It seems that all reviewers' concerns are addressed.

**Reviewer Scores:**

Only one reviewer retains a negative score. The authors have clarified this point in their response, but the reviewer did not reply further.

This remaining concern relates to the core contribution of GMAN. In contrast, the other three reviewers recognize and support the value of this contribution.

---

### Decision · Program_Chairs · 2026-01-26

Accept (Poster)